# Narrow equilibrium window for complex coacervation of tau and RNA under cellular conditions

Yanxian Lin[1†], James McCarty[2†], Jennifer N Rauch[3,4], Kris T Delaney[5], Kenneth S Kosik[3,4], Glenn H Fredrickson[5,6], Joan-Emma Shea[2,7], Songi Han[2,6]*

[1]Biomolecular Science and Engineering, University of California Santa Barbara, Santa Barbara, United States; [2]Department of Chemistry and Biochemistry, University of California Santa Barbara, Santa Barbara, United States; [3]Department of Molecular, Cellular and Developmental Biology, University of California Santa Barbara, Santa Barbara, United States; [4]Neuroscience Research Institute, University of California Santa Barbara, Santa Barbara, United States; [5]Materials Research Laboratory, University of California Santa Barbara, Santa Barbara, United States; [6]Department of Chemical Engineering, University of California Santa Barbara, Santa Barbara, United States; [7]Department of Physics, University of California Santa Barbara, Santa Barbara, United States

*For correspondence:
songi@chem.ucsb.edu

[†]These authors contributed equally to this work

Competing interests: The authors declare that no competing interests exist.

**Abstract** The mechanism that leads to liquid-liquid phase separation (LLPS) of the tau protein, whose pathological aggregation is implicated in neurodegenerative disorders, is not well understood. Establishing a phase diagram that delineates the boundaries of phase co-existence is key to understanding whether LLPS is an equilibrium or intermediate state. We demonstrate that tau and RNA reversibly form complex coacervates. While the equilibrium phase diagram can be fit to an analytical theory, a more advanced model is investigated through field theoretic simulations (FTS) that provided direct insight into the thermodynamic driving forces of tau LLPS. Together, experiment and simulation reveal that tau-RNA LLPS is stable within a narrow equilibrium window near physiological conditions over experimentally tunable parameters including temperature, salt and tau concentrations, and is entropy-driven. Guided by our phase diagram, we show that tau can be driven toward LLPS under live cell coculturing conditions with rationally chosen experimental parameters.
DOI: https://doi.org/10.7554/eLife.42571.001

## Introduction

Protein liquid-liquid phase separation (LLPS) is a process in which proteins assemble and partition into a protein-dense phase and a protein-dilute phase. The proteins in the dense phase form droplets, and retain liquid-like mobility, as shown by NMR measurements (*Burke et al., 2015*; *Brady et al., 2017*). The process of LLPS in vitro has been observed for decades (*Anderson and Kedersha, 2006*; *Brangwynne et al., 2009*; *Wippich et al., 2013*; *Veis and Aranyi, 1960*; *Arneodo et al., 1988*; *Water et al., 2014*), but the field has recently been invigorated by the realization that LLPS also occurs in vivo, suggesting a possible physiological role for these assemblies (*Brangwynne et al., 2009*; *Molliex et al., 2015*; *Hyman et al., 2014*). The overwhelming majority of proteins observed to undergo LLPS are intrinsically disordered proteins (IDPs) (*Uversky et al., 2015*), and much of the research thus far has focused on ALS-related IDPs, including FUS (*Molliex et al., 2015*; *Li et al., 2013*; *Murakami et al., 2015*; *Patel et al., 2015*), hnRNPA2B1

**eLife digest** Proteins make up much of the machinery of cells and perform many roles that are essential for life. Some important proteins – known as intrinsically disordered proteins – lack any stable three-dimensional structure. One such protein, called tau, is best known for its ability to form tangles in the brain, and a buildup of these tangles is a hallmark of Alzheimer's disease and many other dementias.

Tau is also one of a number of proteins that can undergo a process called liquid-liquid phase separation: essentially, a solution of tau separates into a very dilute solution interspersed with droplets of a concentrated tau solution, similar to an oil-water mixture separating into a very watery solution with drops of oil. Understanding the conditions that lead to spontaneous liquid-liquid phase separation might give insight into how the tau tangles form. However, it was not known whether it is possible in principle for liquid-liquid phase separation of tau to occur in a living brain.

Lin, McCarty et al. have now used an advanced computer simulation method together with experiments to map the conditions under which a solution containing tau undergoes liquid-liquid phase separation. Temperature as well as the concentrations of salt and the tau protein all influenced how easily tau droplets formed or dissolved, and the narrow range of conditions that encouraged droplet formation fell within the normal conditions found in the body, also known as "physiological conditions". This suggested that tau droplets might form and dissolve easily in living systems, and possibly in the brain, depending on the precise physiological conditions. To explore this possibility further, tau protein was added to a dish containing living cells. As the map suggested, slightly adjusting temperature or protein concentrations caused tau droplets to form and dissolve, all while the cells remained alive.

The map provided by this study may offer guides to researchers looking for liquid-liquid phase separation in the brain. If liquid-liquid phase separation of tau occurs in living brains, it may be important for determining whether and when damaging tau tangles emerge. For example, the high concentration of tau in droplets might speed up tangle formation. Ultimately, a better understanding of the conditions and mechanism for liquid-liquid phase separation of tau can help researchers understand the role of protein droplet formation in living systems. This may be a process that promotes, or possibly a regulatory mechanism that prevents, the formation of tau tangles associated with dementia.

DOI: https://doi.org/10.7554/eLife.42571.002

and hnRNPA1 (*Kato et al., 2012*), TDP-43 (*Kato et al., 2012*; *Li et al., 2018b*), C9ORF72 (*Kwon et al., 2014*; *Lee et al., 2016*; *Boeynaems et al., 2017*) and Ddx4 (*Nott et al., 2015*). Recently, we and others discovered that another amyloid forming IDP, the microtubule binding protein tau, also undergoes LLPS (*Ambadipudi et al., 2017*; *Zhang et al., 2017*; *Hernández-Vega et al., 2017*; *Ferreon et al., 2018*; *Wegmann et al., 2018*). Interestingly, many of the LLPS forming IDPs have been observed to form amyloid fibrils in cell-free systems (*Murakami et al., 2015*; *Kato et al., 2012*), leading to a number of hypotheses regarding the physiological role of LLPS in regulating aggregation. In particular, a compelling idea is that protein LLPS may be an intermediate regulatory state, which could redissolve into a soluble state or transition to irreversible aggregation/amyloid fibrils (*Murakami et al., 2015*; *Patel et al., 2015*; *Kato et al., 2012*; *Ambadipudi et al., 2017*; *Zhang et al., 2017*).

In a healthy neuron, tau is bound to microtubules. When tau falls off the microtubule under adverse conditions to the cell, tau is solubilized in the intracellular space as an IDP. Under certain conditions, tau forms intracellular fibrillary tangles, a process linked to neurodegenerative tauopathies that include Alzheimer's disease. In recent work, we showed that tau in neurons strongly (nanomolar dissociation constant) and selectively associates with smaller RNA species, most notably tRNA (*Zhang et al., 2017*). We also found tau and RNA, under charge matching conditions, to undergo LLPS (*Zhang et al., 2017*) in a process determined to be complex coacervation (CC) (*Bungenberg de Jong, 1949*). We found that tau-RNA LLPS is reversible, and persisted for >15 hr without subsequent fibrilization of tau, and hypothesized that LLPS is potentially a physiological and regulatory state of tau.

In this work, we characterize the phase diagram of tau-RNA LLPS using a combination of experiment and simulation, and thereby specify the conditions that drive the system toward a homogeneous phase or an LLPS state. We study a N-terminus truncated version of the longest isoform of human 4R tau in vitro, and first demonstrate that tau-RNA complexation is reversible, and that tau remains dynamic and without a persistent structure within the dense phase. The phase coexistence curve separating a supernatant phase from a condensate phase is determined by the system's free energy, which in turn is state dependent, that is dependent on concentration, temperature, salt, and the nature of the interaction strength between the various solution constituents, including the solvent. We construct the phase diagram from cloud-point measurements of the onset of complex coacervation under varying conditions of temperature, salt, and polymer concentrations. These experiments establish the features and phase coexistence boundaries of the phase diagram, which we then model using theory and simulation to rationalize and understand the physical mechanisms that drive and stabilize LLPS.

A number of theoretical models can be used to model LLPS, each with their own advantages and disadvantages. Ideally, one would turn to simulations at atomic resolution in explicit solvent; however, such models are computationally prohibitive given the multiple orders of magnitude in time and length scales involved in LLPS. Turning to the polymer physics literature, theoretical treatments of simplified coarse-grained models are much more computationally tractable, and offer useful insight. Although approximate, analytical theories can be formulated, providing an extremely efficient platform for describing the thermodynamics of polyelectrolyte mixtures (*Sing, 2017*). These include the Flory-Huggins model (*Flory, 1953*), the Voorn-Overbeek model (*Veis and Aranyi, 1960*; *Spruijt et al., 2010*; *Overbeek and Voorn, 1957*; *Tainaka, 1979*; *Veis, 1963*; *Tainaka, 1980*; *Nakajima and Sato, 1972*), the random-phase approximation (*Borue and Erukhimovich, 1988*; *Borue and Erukhimovich, 1990*; *Castelnovo and Joanny, 2001*), the Poisson-Boltzmann cell model (*Biesheuvel and Cohen Stuart, 2004a*; *Biesheuvel and Stuart, 2004b*), as well as other more sophisticated approaches (*Lytle and Sing, 2017*; *Shen and Wang, 2018*; *Shen and Wang, 2017*), which have been applied to synthetic polymers with low sequence heterogeneity (*Spruijt et al., 2010*; *Chollakup et al., 2010*; *Zalusky et al., 2002*; *Li et al., 2018a*; *de la Cruz et al., 1995*), and to proteins with single composition (*Brady et al., 2017*; *Nott et al., 2015*; *Banjade and Rosen, 2014*; *Banjade et al., 2015*). While such models have been successful in describing simpler polyelectrolytes, it is less apparent that these models are suitable to describe the complex coacervation of the more complicated tau-RNA system. The simplest approach that one can use is the Flory-Huggins (FH) model, augmented by the Voorn and Overbeek (VO) correction to describe electrostatic correlations. This model is widely used to model LLPS; however, while experimental data can be fit to the model (*Brady et al., 2017*; *Nott et al., 2015*), ultimately the FH-VO model has serious inadequacies. The original Flory-Huggins model is a mean-field theory, which means that fluctuations in polymer densities away from their average value in each phase are neglected. Augmenting the FH model with a VO treatment of electrostatics approximately accounts for charge correlations, but it entirely neglects chain-connectivity (*Qin and de Pablo, 2016*). Thus, the FH-VO model is unable to model the spatially varying charge distribution along the polymer backbone. Ideally, one would like to introduce chain connectivity, charge correlation, and uneven charge distribution into a more realistic polymer physics model; however, a full treatment of polymer density fluctuations is analytically intractable. One possible approach is to pursue a Gaussian approximation to field fluctuations, also known as the random phase approximation (RPA) (*Kudlay and Olvera de la Cruz, 2004*; *Kudlay et al., 2004*; *Castelnovo and Joanny, 2000*). The RPA model can be viewed as a lowest-order correction to the mean field approximation and was recently introduced to describe the charge pattern and sequence-dependent LLPS of IDPs (*Lin et al., 2017*; *Lin et al., 2016*). The advantage of the RPA model, over the mean-field FH-VO model, is that charge correlations are introduced in a formally consistent manner. Nonetheless, it has been recently demonstrated that the RPA model fails to quantitatively predict polymer concentrations in the dilute phase, given that higher order fluctuations are important in this regime (*Delaney and Fredrickson, 2017*; *Das et al., 2018*).

Of all the models described above, fitting experimental data with the FH or FH-VO theory is currently the preferred methodology in the LLPS community to describe and analyze phase diagrams. We demonstrate that this model can be fit to describe our experimental data, but the learning outcome from this modeling is limited. Thus, we take a different approach by computing the exact phase diagram of an off-lattice coarse-grained polyelectrolyte model using field theoretic

simulations (FTS). FTS is a numerical approach that allows one to fully account for fluctuations, and thus to compute equilibrium properties from a suitably chosen coarse-grained representation of the true system without the need for analytical approximation. The ability to perform field theoretic simulations enables us to include the important physics of polymer sequence-specificity that cannot be captured by FH-VO, including charge distribution and chain connectivity. Results from FTS are compared to those obtained from the FH-VO model.

The model substantiates the experimental phase diagram that the equilibrium window for the complex coacervation of tau and RNA under cellular conditions is narrow. Guided by the phase diagram, empirically obtained from in vitro experiments and validated by simulation, we finally show that LLPS of tau-RNA can be established and rationalized under cellular co-culturing conditions in the presence of live cells.

## Results

### Tau-RNA complex coacervate is reversible and a dynamic liquid phase

Truncated versions of the longest isoform of human 4R tau, residues 255–441 (*Peterson et al., 2008*) and residues 255–368 were used to study tau-RNA complex coacervation (CC). A C291S mutation was introduced to either tau variant, resulting in single-cysteine constructs. Thioflavin T assays and TEM imaging were performed showing these variants retain the capability to form fibrils with morphology similar to full length tau. Unless otherwise specified, we refer to these two single-cysteine tau constructs as tau187 and tau114 (tau114 is close to K18, 244–372 [*Gustke et al., 1994*]), respectively, while tau refers collectively to any of these variants (see Materials and methods for experimental details). Importantly, experiments were performed with freshly eluted tau within 30 min upon purification to minimize the effects of possible disulfide bond formation. This minimizes the influence of the cysteine mutations on the LLPS behavior of tau-RNA CC. The single-cysteine containing tau187 can be singly spin labeled at site 322, referred to as tau187-SL (see Materials and methods). Full length tau, tau187 and tau114 are overall positively charged with an estimated +3, +11 and +11 charge per molecule at neutral pH, respectively, based on their primary sequences. The charged residues of tau are more concentrated in the four repeat domains (*Figure 1A*). PolyU RNA (800 ∼ 1000 kDa), which is a polyanion carrying one negative charge per uracil nucleotide, was used in this study and henceforth referred to as RNA (*Figure 1A*). Under ambient conditions, both tau and RNA are soluble and stable in solution. By mixing tau and RNA under certain conditions, a turbid and milky suspension was obtained within seconds, where tau and RNA formed polymer-rich droplets (dense phase) separated from polymer-depleted supernatants (dilute phase) (*Figure 1B*). These polymer-rich droplets are tau-RNA CCs. We began by determining the concentration of the dense and dilute phases. After mixing and centrifuging 60 μL tau187-RNA droplet suspension, we separated a polymer-rich phase of volume <1 μL with a clear boundary against the dilute supernatant phase. Applying UV-Vis spectroscopy (see Materials and methods), we determined the concentration of tau and RNA inside the droplets as >76 mg/mL and >17 mg/mL with partitioning factors of >15 and >700, respectively. This is consistent with our previously findings that tau is virtually exclusively partitioned within the dense phase (*Zhang et al., 2017*). High-protein concentrations are typically correlated with higher propensity for irreversible protein aggregations. In order to verify that there was indeed no fibril formation, tau187-RNA CCs were prepared by mixing tau187-SL and RNA (see Materials and methods) and monitored by continuous wave electron paramagnetic resonance spectroscopy (For details of cw-EPR experiments see Materials and methods). The cw-EPR spectra shows no broadening (*Figure 1C*), and the cw-EPR spectra analysis reveals an unchanged rotational correlation time for the spin label of tau187-SL, τ, of 437 ± 37 ps as a function of time after >96 hr of incubation at room temperature (*Figure 1D*, turquoise) (see Materials and methods). For comparison, cw-EPR spectra and τ were recorded of tau187-SL alone in buffer, and of tau187-SL in the presence of heparin under fibril forming conditions. Tau187-SL alone in buffer showed cw-EPR spectra overlapping with those of tau187-RNA CC, and rotational correlation time τ, 425 ± 16 ps, nearly identical to the τ of tau187-SL CCs (*Figure 1D*, red). In contrast, tau187-SL with heparin shows a significantly broadened cw-EPR spectrum and an increasing τ to 2.3 ± 0.7 ns (*Figure 1C,D*, green). Note that a hundreds of ps range of τ corresponds to rapid tumbling of the spin label, whose rotational degree of freedom is minimally hindered by molecular associations, while a several ns range of

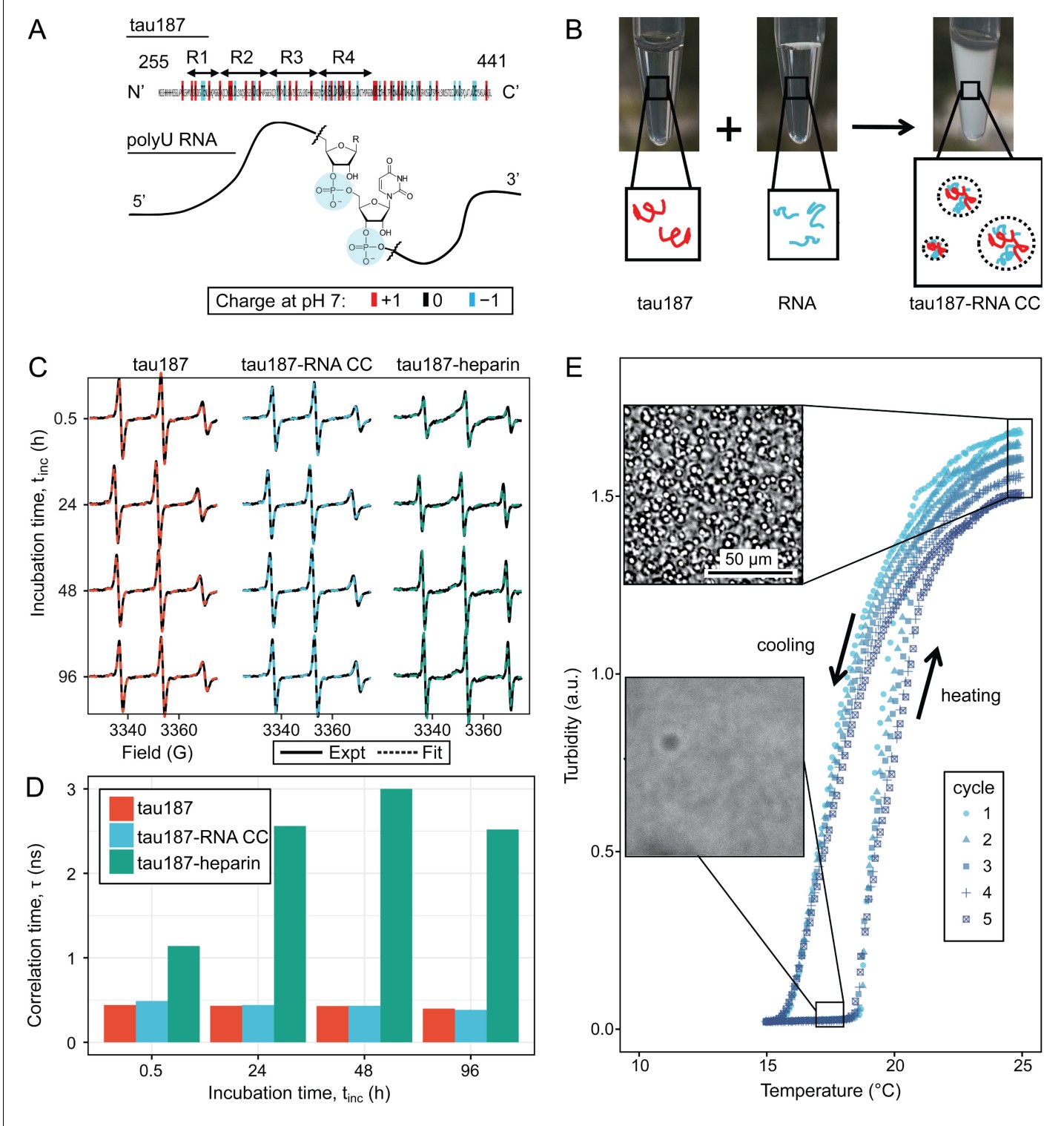

**Figure 1.** Steady tau dynamics and reversible droplet formation of tau-RNA complex coacervates. (**A**) Diagram of tau187 and polyU RNA. Tau187 is a truncated version of full-length human tau (2N4R 255-441) containing repeat domains and C terminal. At neutral pH experimental conditions, tau187 is overall positively charged; while RNA consists of a phosphate backbone and is negatively charged. (**B**) Scheme of tau-RNA CC preparation. Mixing clear tau187 and RNA solutions at proper conditions results in a turbid solution containing liquid droplets. (**C**) X-band cw-EPR spectra (solid line) of tau187 solution (tau187, red), tau187-RNA CC (blue) and tau187-heparin (green) at room temperature with different incubation time, $t_{inc}$. Samples contains 500 μM tau with 20% spin-labelled. EPR simulation were performed (see Materials and methods) and the fitted spectra is shown as a dashed

*Figure 1 continued on next page*

*Figure 1 continued*

line. (D) Rotational correlation time, $\tau_R$ extracted from EPR simulation shown in (C) (see Materials and methods). (E) Turbidity of tau187-RNA suspension in consecutive heating-cooling cycles. Confocal images represented samples at 19 °C and 25 °C. Temperatures were ramped at 1 °C/min.

DOI: https://doi.org/10.7554/eLife.42571.003

The following source data and figure supplements are available for figure 1:

**Source data 1.** Primary sequences of tau187 and tau114 used in the experiments and FH-VO calculation.

DOI: https://doi.org/10.7554/eLife.42571.008

**Figure supplement 1.** ThT fluorescence of tau-RNA CC.

DOI: https://doi.org/10.7554/eLife.42571.004

**Figure supplement 2.** RNase induces turbidity decrease of tau187-RNA CC suspension.

DOI: https://doi.org/10.7554/eLife.42571.005

**Figure supplement 3.** Turbidity of tau-RNA CC at varying charge ratios and ionic strength.

DOI: https://doi.org/10.7554/eLife.42571.006

**Figure supplement 4.** Tau-RNA CC upon addition of heparin.

DOI: https://doi.org/10.7554/eLife.42571.007

$\tau$ corresponds to slow tumbling and molecular hindering by association or confinement. The Thioflavin T (ThT) fluorescence curves of the same sample system as a function of time confirms the absence of amyloid aggregate formation in tau187-RNA CCs (*Figure 1—figure supplement 1*). These results together suggest that tau187-RNA CCs are in an equilibrium state, in which tau retains its solution-like dynamics.

Next, we investigated the reversibility of tau187-RNA complex coacervation. Tau187-RNA CCs were prepared again and incubated by cyclically ramping the temperatures (1°C/min) upwards and downwards, while the absorbance at $\lambda$ = 500 nm was monitored, referred to as turbidity hereafter. Ramping rates of 0.5°C/min and 1°C/min were tested, but the results shown to be indistinguishable. Microscopy images were concurrently acquired at low and high turbidity, confirming the appearance and abundance of CC droplets correlating with turbidity increase, and *vice versa* (*Figure 1E*). The turbidity-temperature curves show that at high temperature, samples became turbid with $Abs_{500}$ ~1.5 and abundance of CCs, while at low temperature, samples became transparent with $Abs_{500}$ ~0 and absence of CCs. This demonstrates tau187-RNA CC formation is favored at higher temperature, following clearly a lower critical solution temperature behavior (LCST) (*Figure 1E*) (*Siow et al., 1972*). By cycling the temperature, we robustly and reversibly changed the tau187-RNA mixture between a turbid state to a completely transparent state (*Figure 1E*). The transition temperatures at which the turbidity emerged during heating and vanished during cooling stay invariant with repeated heating-cooling cycles. The method of extracting a cloud point for the LCST transition temperature from such data will be described in detail in the next section. Importantly, the history of temperature change does not affect the resulting state. Hence the formation and dissolution of tau187-RNA CCs are reversible and consistent with a path-independent equilibrium process. We point out that the maximum turbidity value successively decreases with each heating cycle (*Figure 1E*), even though the transition temperatures remain invariant. This can be attributed to slow degradation of RNA with time, (as demonstrated in *Figure 1—figure supplement 2*) by verifying an altered turbidity change in the presence of RNase or RNase inhibitor.

It is understood that upon gradual heating of the solution phase, the mechanism of LLPS proceeds via a nucleation process (*Berry et al., 2015*), and hence there is a kinetic barrier evidenced by the observed hysteresis in *Figure 1E*. Nonetheless, we conclude that the final tau-RNA CC state reached upon heating is a true thermodynamic state, and thus can be modeled by an equilibrium theory of phase separation.

## Tau-RNA complex coacervate phase diagram

To understand the principles and governing interactions driving tau-RNA CC formation, we constructed a phase diagram for tau187-RNA CC by measuring the transition temperature – to be described in greater detail below – as a function of protein concentration and salt concentration. We first recorded tau187-RNA turbidity at various [tau], [RNA] and [NaCl] values, ranging from 2 to 240 μM, 6–720 μg/mL and 30–120 mM, respectively. Titrating RNA to tau187, the turbidity was found to be peaked when [RNA]:[tau] reached charge matching condition at which the charge ratio between

net positive and negative charges was 1:1 (which for tau187 and RNA used in this study corresponded to [tau187]:[RNA]=1 μM: 3 μg/mL), validating once more that LLPS is driven by complex coacervation (CC) (*Figure 1—figure supplement 3*). Henceforth, all phase diagram data are acquired at a charge matching condition between RNA and tau. Titrating NaCl to tau187-RNA, CC formation showed a steady decrease of turbidity (*Figure 1—figure supplement 3*). Combined, these demonstrate that tau187-RNA CC favors the condition of charge balance and low ionic strength, which is consistent with known properties of CC and previous findings (*Zhang et al., 2017*).

We next investigated the phase separation temperatures under various sample compositions. Tau187-RNA CCs were prepared with a fixed [tau]:[RNA] ratio corresponding to the condition of net charge balance. Therefore, the composition of tau187-RNA CC can be determined by [tau] and [NaCl]. Samples were heated at 1°C/min between T = 15–25°C, while the turbidity was monitored. The turbidity-temperature data of the heating curves were then fit to a sigmoidal function, so that the cloud point temperature, $T_{cp}$, could be extracted as shown in *Figure 2A* ($T_{cp}$ was determined from heating curves out of practical utility; $T_{cp}$ from cooling curves is possibly closer to thermodynamic transitions). The experimental cloud-point temperature $T_{cp}$ for CC formation as a function of [tau] and [NaCl] are shown (as points) in *Figure 2B* and *Figure 2C*. The experimental data points show that increasing [tau] lowers $T_{cp}$, favoring CC formation, while increasing [NaCl] raises $T_{cp}$, disfavoring CC formation. Such trends were observed at two [NaCl] and two [tau] values, respectively (*Figure 2B and C*). Experimentally, $T_{cp}$ was determined for a range of [tau] and [NaCl] conditions (see *Figure 2—figure supplement 1*). We point out that there is certain level of variability in the observed $T_{cp}$, which can result from pH fluctuation of the ammonium acetate buffer upon tau-RNA addition, as well as RNA degradation as demonstrated in *Figure 1—figure supplement 2*.

The features of the Tau-RNA CC phase diagram were also investigated by comparing tau187 and tau114. Tau187-RNA CC and tau114-RNA CC were prepared with 20 μM tau187 and 28 μM tau114, so that the total concentration of polymer, that is tau and RNA, reaches 0.5 mg/mL. Turbidity was recorded at varying [NaCl]. Similar to the observation with tau187-RNA CC, tau114-RNA CC showed decreasing turbidity at increasing [NaCl] (*Figure 2—figure supplement 2*). The [NaCl] values where turbidity reaches 0 were estimated as 131 mM and 150 mM for tau187 and tau114, respectively, implying CC formation is more favorable with tau114 that hence can sustain higher [NaCl]. Based on this, 20 μM of tau187, 131 mM of NaCl and room temperature, 20°C, were used as the phase separation conditions ([tau], [NaCl] and $T_{cp}$) for tau187, and 28 μM, 150 mM and 20°C for tau114. These two experimental conditions were used in the next section for comparing the two constructs of tau.

## Flory-Huggins-Voorn-Overbeek fit to experimental phase diagram

We next used the FH-VO model to fit the experimental data for the tau187-RNA CC system, as is commonly done in LLPS studies. Despite its theoretical deficiencies, the FH-VO model is commonly used for its simplicity and ease of implementation. Our system consists of five species: tau187, RNA, monovalent cation ($Na^+$), anion ($Cl^-$) and water. For simplicity, we explicitly consider only the effect of excess salt, and do not include polymer counterions. The FH-VO model maps these five species onto a three-dimensional lattice (*Figure 2D*). Each polymer is treated as a uniform chain with degree of polymerization N and average charge per monomer σ. N was taken as the average chain length of the species (one for monovalent ions). The charge density σ of RNA, monovalent ions and water were set to 1, 1 and 0, respectively. The values for σ of tau187 or tau114 were calculated from the net charge at neutral pH divided by the chain length. The composition of the species is expressed in terms of the volume fraction ϕ of the occupied lattice sites, which are proportional to the molar concentrations (see Materials and methods for details). As in experiments, tau187-RNA CCs were prepared at fixed [tau]:[RNA] and [$Na^+$]:[$Cl^-$] ratios. Under these two constraints, the volume fraction of all five species in tau187-RNA CC listed above can be determined with two variables, [tau] and [NaCl], which are experimentally measurable.

Given N, σ, [tau], [NaCl] and $T_{cp}$, the task is to find $\phi_{tau}^I$ and $\phi_{tau}^{II}$, the volume fractions of tau in the dilute and dense, coacervate, phase at equilibrium, that is the binodal coexistence points. The model and procedure is described in detail in the Materials and methods. For each experimental observation of $T_{cp}$ determined for a given [tau] and [NaCl] (*Figure 2—figure supplement 1*), the FH-VO expression has one unknown parameter, the Flory-Huggins χ term. The Flory-Huggins χ parameter is introduced as an energetic cost to having an adjacent lattice site to a polymer segment occupied by a solvent molecule (*Brangwynne et al., 2015*). Here, we take χ to be an adjustable

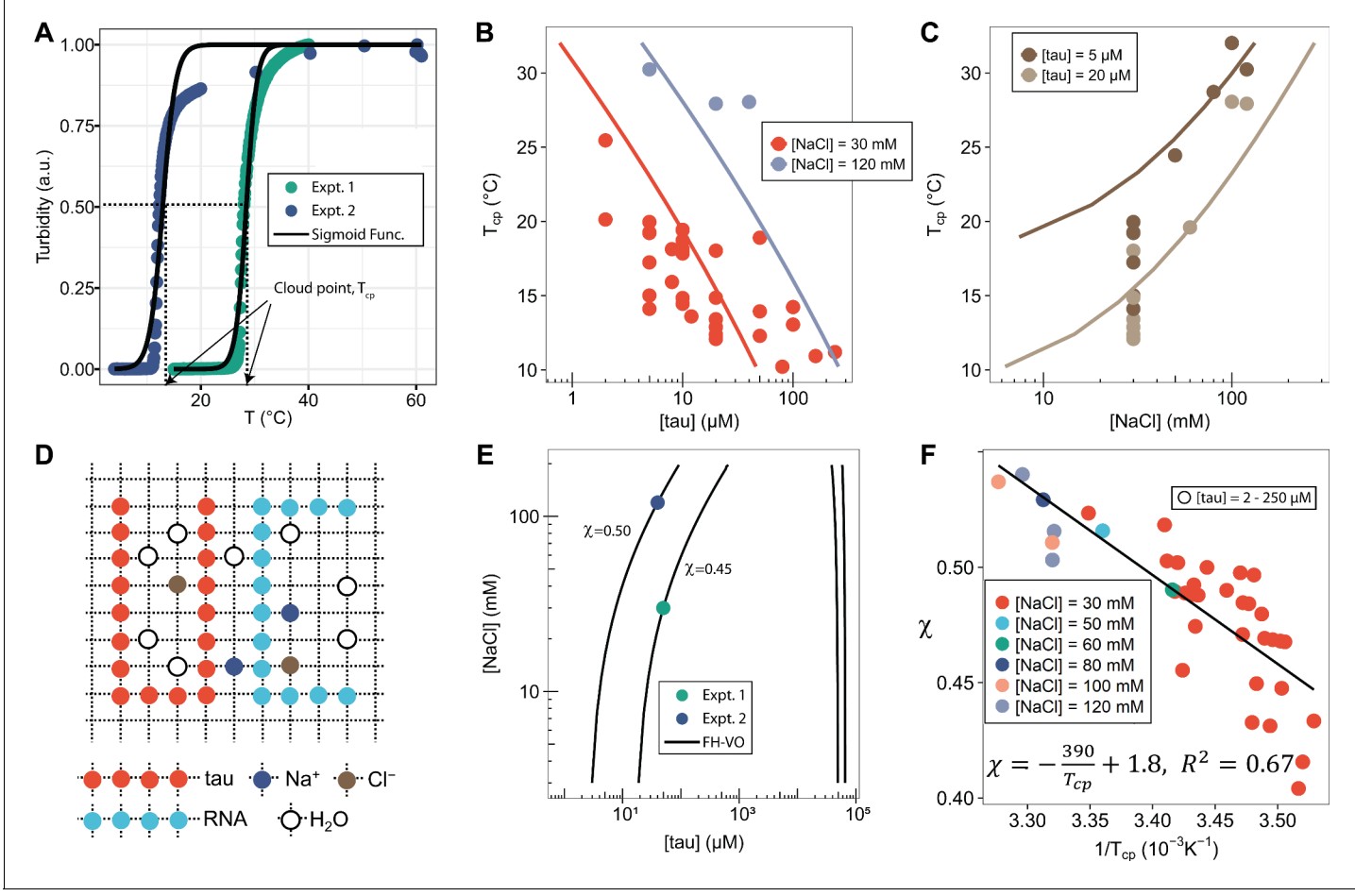

**Figure 2.** FH-VO modeling of tau-RNA complex coacervation. (**A**). Turbidity of tau187-RNA CC upon heating (Expt.1 ([tau], [NaCl]) = (50 μM, 120 mM), green dots; Expt.2 ([tau], [NaCl]) = (40 μM, 30 mM), purple dots). Absorbance at λ = 500 nm were normalized and used as turbidity value. Turbidity-temperature data of the heating curves were fitted with a sigmoidal function (solid line) as described in (Materials and methods), and the temperature at which normalized turbidity reaches 0.5 was assigned to cloud point, $T_{cp}$. (**B**)-(**C**). Experimental phase diagram (points) showing [tau] vs. $T_{cp}$ and [NaCl] vs. $T_{cp}$ along with the binodal curve generated from fitting the data to the FH-VO model with χ = χ($T_{cp}$) (solid line). (**D**). Diagram of Flory Huggins lattice. Tau and RNA are represented by consecutively occupied lattice sites. (**E**). Each experimental condition in (**A**) was independently fit to the FH-VO model (solid lines) to obtain an empirical χ value. (**E**) shows two representative curves. These empirically determined values of χ are shown as points in (**F**). The solid line in (**F**) is a linear regression, generating χ = χ($T_{cp}$), which is then used to generate the binodal lines in (**B**) and (**C**).

DOI: https://doi.org/10.7554/eLife.42571.009

The following source data and figure supplements are available for figure 2:

**Source data 1.** Degree of polymerization and average charge densities of species used in the FH-VO calculation.
DOI: https://doi.org/10.7554/eLife.42571.013
**Source data 2.** Thermodynamics calculated by FH-VO at 300 K and physiological relevant salt concentrations.
DOI: https://doi.org/10.7554/eLife.42571.014
**Figure supplement 1.** Turbidity-temperature data and cloud points determinations of various [tau] and [NaCl].
DOI: https://doi.org/10.7554/eLife.42571.010
**Figure supplement 2.** Phase diagrams of tau114 vs tau187.
DOI: https://doi.org/10.7554/eLife.42571.011
**Figure supplement 3.** Full phase diagram of tau187-RNA CC.
DOI: https://doi.org/10.7554/eLife.42571.012

parameter, such that given a suitable expression for χ, the complete binodal curve can be modeled with the FH-VO theory. Consequently, we first solved for χ at each given experimental condition, so that the theoretical binodal curve intersects the experimental data point. *Figure 2E* shows two representative examples of a theoretical binodal curve (solid line) intersecting a single experimental

data point at the given [NaCl] and [tau]. This procedure gives an empirical $\chi$ parameter for each experimental data point, as collated in *Figure 2F* as a function of $1/T_{cp}$. We then performed to this set of experimental data a least-squares fit of the empirical $\chi$ parameter to the form $A + B/T$ (*Figure 2F*), yielding an expression of the temperature dependence of $\chi$ of

$$\chi(T) = 1.8 - \frac{390}{T}, \ R^2 = 0.67 \tag{1}$$

A temperature dependence of $\chi$ in the form of *Equation 1* (consistent with the observed LCST), can originate from hydrophobic interactions between non-polar groups, whose interaction strength tends to increase with temperature (*Lin et al., 2018*; *Dias and Chan, 2014*). This explanation has also been used to describe cold denaturation of proteins (*Dill et al., 1989*).

Finally, from this expression for $\chi(T)$, we computed the binodal curves that establishes the phase coexistence as a function of $T_{cp}$, [tau] and [NaCl], shown as solid lines, only for the dilute phase coexistence for $T_{cp}$ vs [tau] (*Figure 2B*) and $T_{cp}$ vs [NaCl] (*Figure 2C*). For the full phase diagram showing both dilute and dense binodal curves see *Figure 2—figure supplement 3*. The experimental data (shown as points) and computed binodal curves both exhibited a decreasing $T_{cp}$ with increasing [tau] and an increasing $T_{cp}$ with increasing [NaCl]. This simply establishes that tau-RNA CC favors higher tau concentrations in the 1–240 µM range and lower ionic strength in the 30–120 mM range tested here.

Binodal curves for tau114-RNA CC were also computed and are compared with tau187-RNA CC, along with experimental data (*Figure 2—figure supplement 2*). Comparison of the two constructs shows that tau114-RNA CC has a lower $T_{cp}$ than tau187-RNA CC, suggesting it is more favorable to phase separation. This qualitatively agrees with experimental observations. Notice that the shorter tau114 has a slightly higher propensity to form CC as compared to the longer tau187 fragment, an observation that is opposite of what one would expect from purely entropic considerations based on the mixing of homopolymers or simple coacervation. One possible explanation could be the increased charge density of tau114 with respect to tau187, indicating the importance of both charge sequence and charge density for the phase diagram. Additional short-ranged sequence-specific interactions between tau114 and RNA that are not present in tau187 is another possibility that is not considered in the present model.

## Field theoretic simulations of a coarse-grained model of tau-RNA complex coacervation

Although the FH-VO model can be brought into agreement with experiment through a judicious choice of $\chi$, it is fundamentally unsound from a theoretical perspective, noticeably because it neglects connectivity between charges on the same chain. This is a severe limitation because it is expected that subtle difference in primary amino acid sequences may have a profound effect on the phase diagram. A particularly appealing alternative to gain insights into the thermodynamics of LLPS is to perform field theoretic simulations (FTS) on a physically motivated polyelectrolyte model (*Figure 3*), in which each amino acid is represented by a single monomeric unit of length $b$ in a coarse-grained bead-spring polymer model. The charge of each segment is unambiguously assigned from the particular amino acid charge at pH 7.0. In addition to harmonic bonds between nearest neighbors, which enforces chain connectivity, all segment pairs interact via two types of non-bonded potentials: a short-ranged excluded volume repulsion and a long-range electrostatic interaction between charged monomers (see *Figure 3*). We take the polymers to be in a slightly good solvent, meaning that favorable interactions between monomers and solvent cause chain swelling. In such cases, the excluded volume interaction is modeled as a repulsive Gaussian function between all monomer pairs with a strength that increases with solvent quality (*Doi and Edwards, 1988*). Conversely, as solvent quality decreases, the excluded volume repulsion decreases, approaching zero at the so-called theta condition. In the present case, we limit ourselves to the case where the excluded volume is positive and small, that is a good solvent near the theta condition. Simulations are performed using a single excluded volume strength, $v$, identical for all monomers, which is an input parameter in the model and can be adjusted to parameterize the favorable monomer-solvent interactions. Additionally, the long-range electrostatic interactions are described by a Coulomb potential in a screened, uniform, dielectric background. The length scale of the electrostatic interactions is

parameterized by the Bjerrum length $l_B$, which is the distance at which the electrostatic interactions become comparable to the thermal energy $k_BT$ and is defined as

$$l_B = \frac{e^2}{4\pi\epsilon_0\epsilon_r k_B T} \tag{2}$$

where $e$ is the unit of electronic charge, $\epsilon_r$ the dielectric constant ($\epsilon_r = 80$ for water), and $\epsilon_0$ the vacuum permittivity.

The main features of the model used for FTS here are the inclusion of chain connectivity, charge sequence-dependence for the electrostatic interactions based on the primary amino acid sequence of tau, solvation effects which are parameterized by the single excluded volume parameter, $v$, and an electrostatic strength parameterized by the Bjerrum length, $l_B$. FTS is performed in implicit solvent with a uniform dielectric background. We assume that the polymer chains are in a fully dissociated state, and we do not explicitly represent counter ions. The effect of excess salt is included in our model by introducing point charges explicitly, which engage in Coulomb interactions with all other charged species and repel other ions and polymer segments at short distances by the same Gaussian excluded volume repulsion. By introducing explicit small ions in this manner, we are neglecting strong correlations such as counter ion condensation; however, we are allowing for weak correlations of the Debye-Hückel type. The explicit addition of salt will serve to screen the electrostatic interactions and inhibit the driving force for CC, in agreement with the experiments.

Details of the FTS protocol are described in the Materials and methods. By performing FTS at various state points and computing equilibrium properties, we first set out to fully explore the parameter space relevant for LLPS in this model. This involves running simulations at different conditions analogous to experiments. For each simulation, the thermodynamic state of the system is determined by specifying a particular value for the dimensionless excluded volume parameter $v/b^3$, the dimensionless Bjerrum length $l_B/b$, and the dimensionless monomer number density $\rho b^3$. *Figure 4* shows the final polymer density configuration for two representative simulations at a monomer density of $\rho b^3 = 0.22$ at different thermodynamic conditions (see caption for *Figure 4* for details). Although the bulk density is fixed and identical for the two cases, the local polymer density is free to fluctuate. The left simulation box (*Figure 4*) shows a case where a single phase is favored, indicated by a nearly homogenous polymer density throughout the simulation box (white/blue). This is contrasted by the right simulation box (*Figure 4*) depicting the case where the system phase separates into a dilute polymer-deplete region (white) and a dense polymer-rich droplet region (red)—the coacervate phase with the color signifying the polymer density. *Figure 4* shows that given suitable

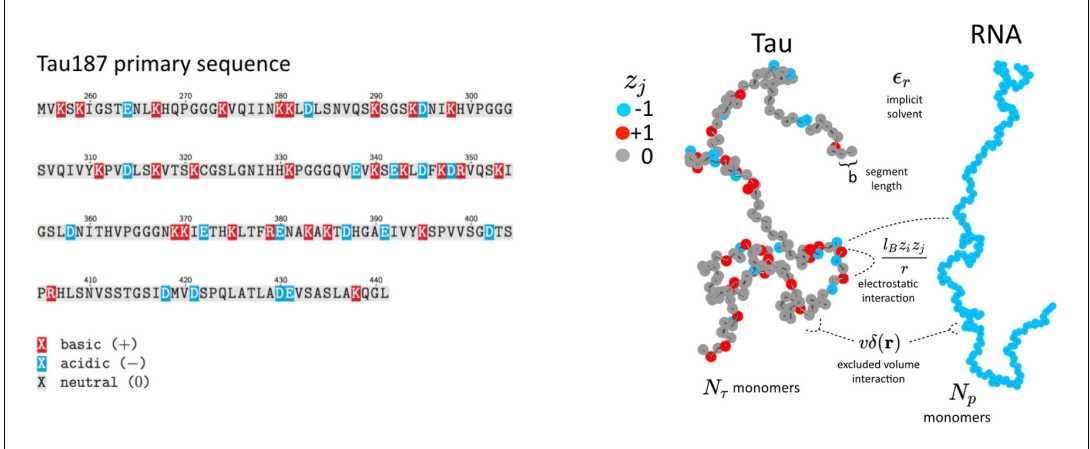

**Figure 3.** Schematic depiction of the tau and RNA polyelectrolyte models. Tau and RNA molecules are represented as bead-spring polymers with segment length $b$ in implicit solvent. Tau is modeled as a polyampholyte with the charge of each monomer determined from the amino acid charge at pH = 7. RNA is modeled as a fully charged polyelectrolyte. In addition to chain connectivity, all monomers interact with an excluded volume repulsive potential, and charged monomers interact with a long-ranged Coulomb potential.
DOI: https://doi.org/10.7554/eLife.42571.015

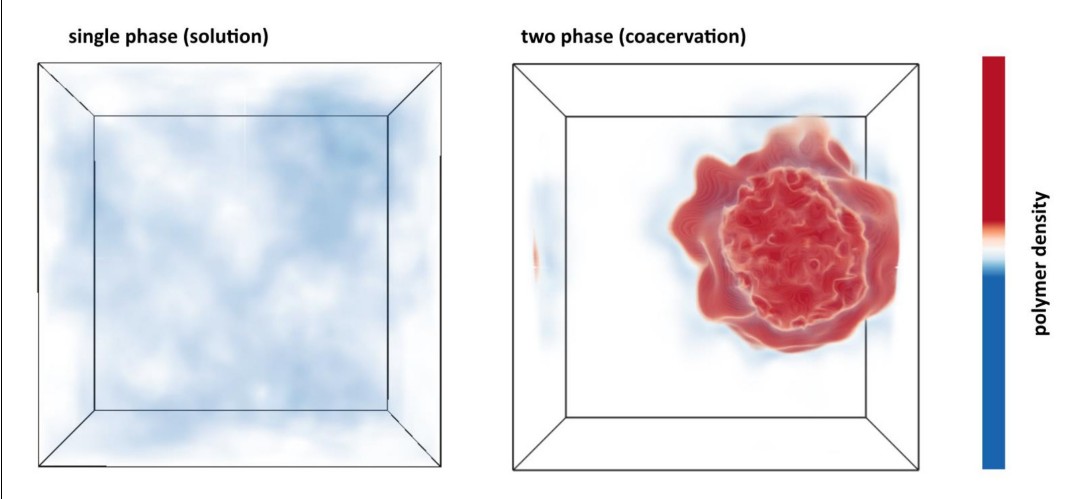

**Figure 4.** Polymer density field of single phase solution and two phase coacervate from FTS. Left: Polymer density profile showing a single solution phase for the condition of relatively weak electrostatic strength $l_B = 0.16\ b$ and relatively high excluded volume (good solvent conditions) $v = 0.02\ b^3$. The solution phase is characterized by near homogeneous low polymer density (white/light blue) throughout the entire simulation box. Right: Polymer density profile showing complex coacervation upon increasing the electrostatic strength to $l_B = 3.25\ b$ and decreasing the solvent quality by lowering the excluded volume to $v = 0.0068\ b^3$. The two phase region is characterized by a distinct region of high polymer density (dark red) and a surrounding region of low polymer density (white) within the same simulation box. The total polymer concentration is the same in both simulations.
DOI: https://doi.org/10.7554/eLife.42571.016

parameterization, FTS can be used to study complex coacervation of a coarse-grained tau-RNA model. Given this observation, we next map out the full phase diagram in the parameter space of the model while fixing the physical parameters of charge sequence, chain length, and chain volume fractions that are consistent with the experimental conditions.

## Field theoretic simulations predict phase equilibria around physiological conditions

The parameters to be explored in connection with phase behavior are the strength of the interactions in the polyelectrolyte model: the excluded volume strength $v$ and the Bjerrum length $l_B$. A direct comparison between FTS and the experimental phase diagram will be deferred until the following section. The phase coexistence points (binodal conditions) for a given value of the excluded volume $v$ and Bjerrum length $l_B$ can be obtained by running many simulations over a range of concentrations, and finding the concentration values at which the chemical potential and the osmotic pressure are equal in both phases (see *Figure 5—figure supplement 1*). The procedure is described in the SI and is repeated for many different $v$ and $l_B$ combinations. The resulting phase diagram will be a three-dimensional surface which is a function of $\rho$, $v$, and $l_B$. In *Figure 5A*, we show a slice of this surface along the $l_B - \rho$ plane with a fixed value of $v = 0.0068\ b^3$, and in *Figure 5B* we show a slice along the $v - \rho$ plane with a fixed $l_B = 1.79\ b$ (at $T = 293$ K, *Equation 3*). It should be noted that *Figure 5* presents the first complete phase diagrams from FTS presented in the literature of a theoretical model describing a biological complex coacervate system. From *Figure 5A*, one can see that $l_B$ and $v$ have counteracting effects, namely increasing $v$ that is caused by increased solvent quality destabilizes the coacervate phase and favors the single phase, whereas increasing $l_B$ that is caused by reduced electrostatic screening favors coacervation, and destabilizes the single phase. The physical interpretation of the trends in *Figure 5* is that the actual binodal for the experimental system will depend on two competing features: the solvent quality proportional to $v$, which inhibits coacervation, and the electrostatic strength of the media proportional to $l_B$ which promotes coacervation.

The FTS-derived phase diagram shown in *Figure 5* provides a guide how to experimentally tune the window for complex coacervation by changing the relative contribution of the solvent quality or the dielectric strength. Experimentally, the solvent quality can be decreased by adding crowding agents or by changing the hydrophilic/hydrophobic amino acid composition, while the electrostatic

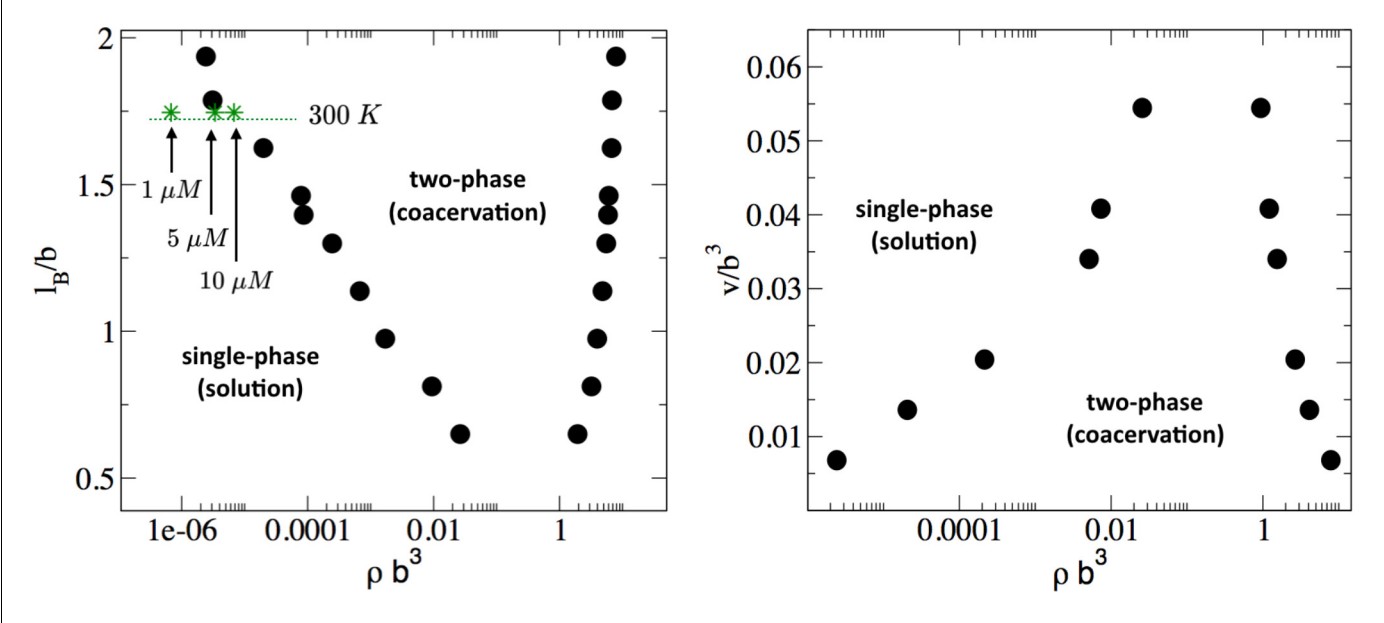

**Figure 5.** Phase diagram of the tau-RNA model obtained from FTS. Left: Binodal points as a function of the Bjerrum length at fixed excluded volume of $v = 0.0068\ b^3$. For comparison three concentrations at 300 K are indicated (arrows) assuming $\epsilon_r = 80$ and $b = 4\ \text{Å}$. Right: Binodal points as a function of the excluded volume at fixed Bjerrum length $l_B = 1.79\ b$.

DOI: https://doi.org/10.7554/eLife.42571.017

The following figure supplement is available for figure 5:

**Figure supplement 1.** Determination of phase coexistence points from FTS.

DOI: https://doi.org/10.7554/eLife.42571.018

strength can be controlled by the salt concentration. Increasing salt concentration tends to decrease the bare electrostatic strength by screening the charges, and this is predicted to stabilize the single phase solution mixture against coacervation, in agreement with experimental observation. We explore these ideas further below in the context of tau coacervation in vivo.

Despite the simplicity of the coarse-grained description, the model predicts that these two competing parameters, excluded volume vs. electrostatic interactions, are nearly balanced around physiological salt concentration, temperature, and protein concentration. Assuming that the relative dielectric constant for water is $\epsilon_r = 80$, and that the segment size $b$ is approximately equivalent to the distance between $C_\alpha$ carbons, that is b~4Å, it follows that $l_B = 1.75b$ at 300 K. ($l_B = 0.7\ nm$ at 300 K). In the $l_B - \rho$ plane (shown in *Figure 5A*), at the cross-section of $l_B = 1.75$, three points for $\rho b^3$ are indicated that correspond to 1, 5, and 10 $\mu M$ for tau concentrations at 300 K. Here, we have implicitly assumed that at physiological temperature and in a crowded cellular environment tau is near the theta condition, and thus $v$ is small. This analysis suggests that small modulation in the experimental conditions, such as changes in the temperature or salt concentration, local pH or crowding effects (via the excluded volume parameter $v$) can readily and reversibly induce complex coacervation in vivo under physiological conditions.

## Comparison between simulation and experiment

In the preceding section, we presented the phase diagram from FTS explicitly in terms of the model parameters of the excluded volume $v$ and Bjerrum length $l_B$. We now seek to compare our simulation results directly with the experimental phase diagram. This requires knowing precisely how the model parameters depend on temperature. We again take the monomer size $b$ to be approximately the distance between the $C_\alpha$ carbons b~4 , and assume a dielectric constant of $\epsilon_w = 80$ for pure water. Although $\epsilon_r$ will depend on temperature, for simplicity, we treat this parameter as a constant such

that the Bjerrum length $l_B \sim 1/T$. Thus, $l_B$ can be estimated at the experimental cloud point temperature directly from *Equation 2* (*Figure 6A*), which leaves only one unknown parameter *v*.

The excluded volume parameter *v* can be related to the residue-residue non-Coulombic interaction potential as

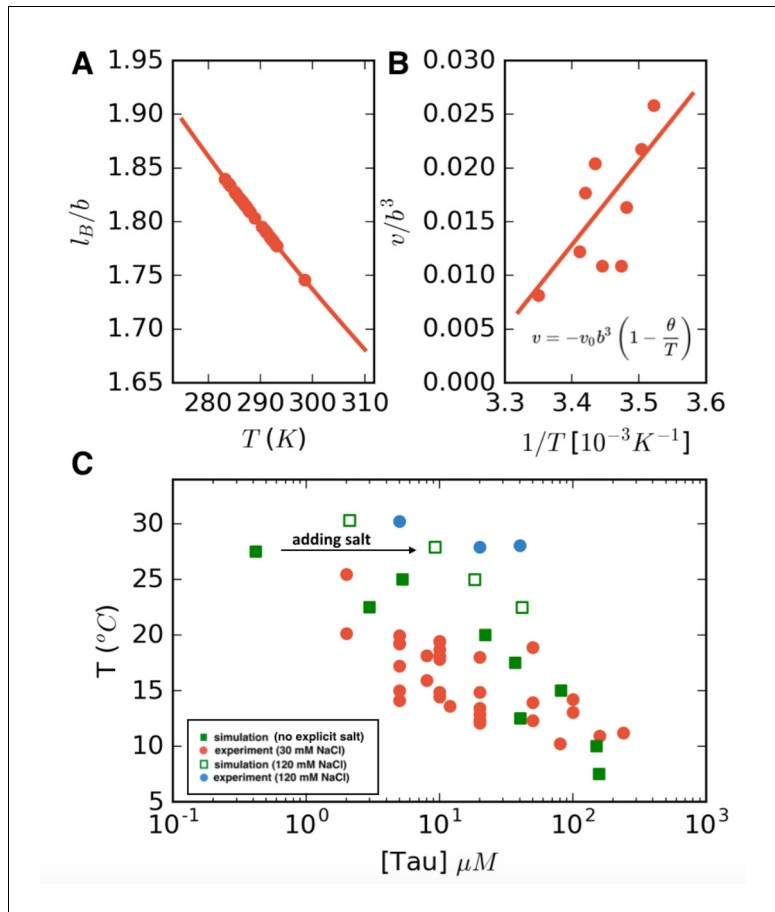

**Figure 6.** LCST phase behavior recapitulated from FTS. (**A**) Temperature dependence of the reduced Bjerrum length ($\varepsilon_r = 80$ for water) shown in red. (**B**) Temperature dependence of the excluded volume obtained by adjusting the excluded volume parameter until FTS agrees with a subset of the experimental data (points). The solid line shows a linear fit to the data which is used to obtain the temperature dependent excluded volume for subsequent simulations. (**C**) FTS coexistence points (filled green squares) obtained by using a temperature dependent Bjerrum length (**A**) and excluded volume (**B**). Experiments performed at 20 mM NaCl are shown in red for comparison. Upon introducing excess salt ions in FTS with a fixed concentration of [NaCl] = 120 mM, the binodal shifts upwards (open green squares). For comparison experiments performed at 120 mM NaCl are shown in blue.

DOI: https://doi.org/10.7554/eLife.42571.019

The following figure supplements are available for figure 6:

**Figure supplement 1.** Full phase diagram of tau187-RNA CC from FTS.
DOI: https://doi.org/10.7554/eLife.42571.020

**Figure supplement 2.** Tau in vitro phosphorylation.
DOI: https://doi.org/10.7554/eLife.42571.021

**Figure supplement 3.** Phosphorylation on tau-RNA complex coacervation.
DOI: https://doi.org/10.7554/eLife.42571.022

**Figure supplement 4.** Concentration of protein in the solution phase and coacervate phase predicted from FTS simulation performed at 20°C on Tau (blue) and P-tau (green) in excess salt.
DOI: https://doi.org/10.7554/eLife.42571.023

$$\nu = \int \left[1 - e^{-\frac{U(r)}{k_B T}}\right] \mathrm{d}^3 r \tag{3}$$

(*Equation 3*) and is typically taken to be proportional to $(1 - \theta/T)$ where $\theta$ is the theta temperature, the temperature at which the chain follows ideal chain statistics (*Doi and Edwards, 1988*; *Stockmayer, 1955*; *Rubinstein and Colby, 2003*). For LCST behavior, it is customary to introduce the form $\nu = -\nu_0(1 - \theta/T)$ where $\nu_0$ controls the magnitude of the excluded volume interactions (*Suzuki et al., 1982*). This form of the excluded volume implies that at temperatures lower than the theta temperature, the excluded volume is repulsive ($\nu > 0$, meaning a good solvent) and for temperatures above the theta point, the excluded volume becomes attractive ($\nu < 0$, poor solvent conditions). By adjusting the excluded volume in FTS to fit a subset of the experimental data, (shown in *Figure 6B*), we then perform a linear fit to obtain a value of $\nu_0 = 0.25b^3$ and $\theta = 309$ K. Note that in the range of temperatures considered, the excluded volume remains positive. However, for temperatures higher than $\theta$, when the excluded volume becomes negative, the polymer chain will collapse, consistent with the observation in the literature (*Bianconi et al., 2012*), which showed that tau undergoes a thermal compaction at high temperatures due to entropic factors (*Stockmayer, 1955*) In such high-temperature regimes, a more sophisticated treatment is needed; however, all our experimental conditions remain below this threshold. Having mapped the two model parameters $\nu$ and $l_B$ to the experimental temperature, we can compare directly the FTS with the experimental results (*Figure 6C*). The calculated FTS data points under the condition of low-salt concentration are shown as filled green squares in *Figure 6C*.

Next, explicit salt ions were introduced as point charges to simulate an excess salt concentration of 120 mM. We make the assumption that the salt is equally partitioned in both phases, and thus the concentration of salt is a constant, allowing us to sweep the polymer concentration at fixed salt concentration to find the phase coexistence points. A more detailed FTS study of salt partitioning performed using a Gibbs ensemble method found that under conditions of nearly charge-balanced polymers, as is the case in the system of this study, the salts are nearly equipartitioned and counterion condensation is not a dominant factor (private communication with Danielsen SPO, McCarty J, Shea J-E, Delaney KT, Fredrickson GH on the "Small ion effects on Self-Coacervation phenomena in block polyampholytes"). Simulations performed in this manner with explicit salt are shown as open green squares in *Figure 6C*. The FTS data clearly demonstrate that the effect of added salt is to stabilize the single solution phase, and to raise the binodal closer to physiological temperature 37°C, in agreement with experiments (filled red and blue circles *Figure 6C*). The complementary dense branch of the binodal curve is also predicted from FTS and is shown in *Figure 6—figure supplement 1*.

## Application to tau phosphorylation

To demonstrate that FTS of tau-RNA LLPS can be applied to consider the effect of post translational charge modification of tau, we tested FTS of phosphorylation as an example. We first phosphorylated tau187 in vitro using mouse brain extract, and confirmed the occurrence of phosphorylation using SDS-PAGE and western blot analysis (see Materials and methods and *Figure 6—figure supplement 2* for details). Next, we titrated phosphorylated tau187 (P-tau) and non-phosphorylated tau187 (Tau) with RNA, while recording turbidity values of the sample. Results showed that after phosphorylation the optimal droplet amount emerged at a lower RNA concentration, as supported by both turbidity and microscopy (*Figure 6—figure supplement 3A,B*). Assuming P-tau-RNA LLPS also follows complex coacervation, we can estimate from the results > 5 additional negative charges are added to P-tau, which is consistent with the fact that phosphorylation adds additional negative charges to protein.

At optimal droplet conditions, we further prepared P-tau-RNA and Tau-RNA samples using 20 μM P-tau with 50 μg/mL RNA and 20 μM Tau with 100 μg/mL RNA, respectively. We titrated the two samples with NaCl, while monitoring turbidity. Results showed the droplet amount in P-tau-RNA sample vanishes at [NaCl]~80 mM, while in Tau-RNA sample vanishes at [NaCl]~260 mM (*Figure 6—figure supplement 3C,D*). These results demonstrated phosphorylation reduces the propensity of tau-RNA LLPS.

For qualitative comparison, we performed FTS on our parameterized tau187 model at equivalent temperature and concentration assuming complete phosphorylation of serine residues S262, S396, S404, S416, and S422, by giving them a charge of $-2$ in the FTS model. Phosphorylated serines were identified according to the literature (*Mair et al., 2016*). For simplicity we assumed a fully phosphorylated tau, which serves as a limiting case, recognizing that in reality, tau will be phosphorylated with some level of variability. To maintain charge neutrality in the simulation box, we add excess salt concentration, so that the total concentration of small ions is estimated to be ~292 mM. *Figure 6—figure supplement 4* shows that added phosphates increases the concentration of tau in the solution phase and decreases tau concentration in the coacervate phase (narrowing the two phase window), which is consistent to the narrowed two phase window observed in *Figure 6—figure supplement 3C*. This is likely due to electrostatic repulsion between the phosphorylated serines and the negatively charged RNA. In contrast, hyper-phosphorylated full-length tau has been shown to favor simple coacervation (*Wegmann et al., 2018*). In full-length tau, most of the phosphorylated sites are not in the positively charged repeat domain region, indicating that its LLPS might follow the same principles as the self-coacervation seen in polyampholytes (*Delaney and Fredrickson, 2017*).

## Application to cell-complex coacervate co-culture

Looking at the experimental and calculated phase diagrams (*Figure 2B and C*), it is seen that under physiological conditions ($T_{cp}$ ~37°C, [NaCl]~100 mM) it is principally feasible for cells to tune the formation of tau-RNA CCs. This has important implications for studying the physiological roles of tau-RNA CCs, and thus we asked if tau-RNA CCs could indeed exist in a biologically relevant media in the presence of living cells. Both the FH-VO theory and FTS predict that the conditions of high-protein concentration, low ionic strength, high temperature and high crowding reagents (leading to solution conditions with a lower effective excluded volume parameter to model the poorer solvent environment in an implicit solvent model [*Jeon et al., 2016*]) would independently favor tau-RNA CC formation. Using these tuning parameters as a guide, we designed several experiments to test the ability for tau-RNA CCs to form in a co-culture with H4 neuroglioma cells. We incubated H4 cells with tau187/tau114-RNA under CC conditions at varying temperatures, polymer concentrations and crowding reagent concentrations. At low polymer concentrations (10 μM tau, 30 μg/ml RNA) no LLPS was observed in the cellular media (*Figure 7*, first column), where increasing the temperature to 37°C did not apparently influence the solution phase (*Figure 7*, first column, first and third row). However, when tau and RNA concentrations were increased (100 μM tau, 300 μg/ml RNA) LLPS could be observed (*Figure 7*, second column). Further, LLPS could also be achieved by adding an additional crowding reagent (here PEG) to low concentration samples of tau and RNA (*Figure 7*, third column). As predicted, LLPS of tau-RNA CC was modulated by (i) temperature, (ii) tau and RNA concentration and/or (iii) the presence of crowding reagent PEG (*Figure 7*). Lowering the temperature to 18°C significantly reduced the number and size of fluorescent droplets, demonstrating that tau-RNA LLPS is indeed tunable by temperature, and demonstrate the biological consequence of the LCST behavior (*Figure 7*, first and third row). These results were consistently found for both tau187 and tau114 systems. The successful application of FTS for tuning and predicting tau-RNA CCs in cellular media is a first step toward understanding the physiological condition under which tau-RNA LLPS, which follows the CC mechanism, can occur. Notice that our truncated tau construct has been demonstrated to undergo LLPS at similar conditions ([tau], [RNA], [NaCl] and temperature) compared with full length tau, 2N4R, in vitro (*Zhang et al., 2017*). The conditions described for LLPS here suggests that conditions exist in vivo under which LLPS by complex coacervation may be achieved by biological regulation mechanisms, and under conditions where tau and the LLPS forming constituents are available in the cytoplasm.

## Discussion

The ability of tau to undergo LLPS via a mechanism of complex coacervation has been recognized in a number of recent publications (*Ambadipudi et al., 2017*; *Wegmann et al., 2018*; *Zhang et al., 2017*). However, to date, the criteria and physical parameters (specifically, polymer concentration, ionic strength, temperature and crowding reagents) that drive tau-RNA CC has not been rationalized. In this paper, we mapped out the experimental phase diagram for tau-RNA CC, and used

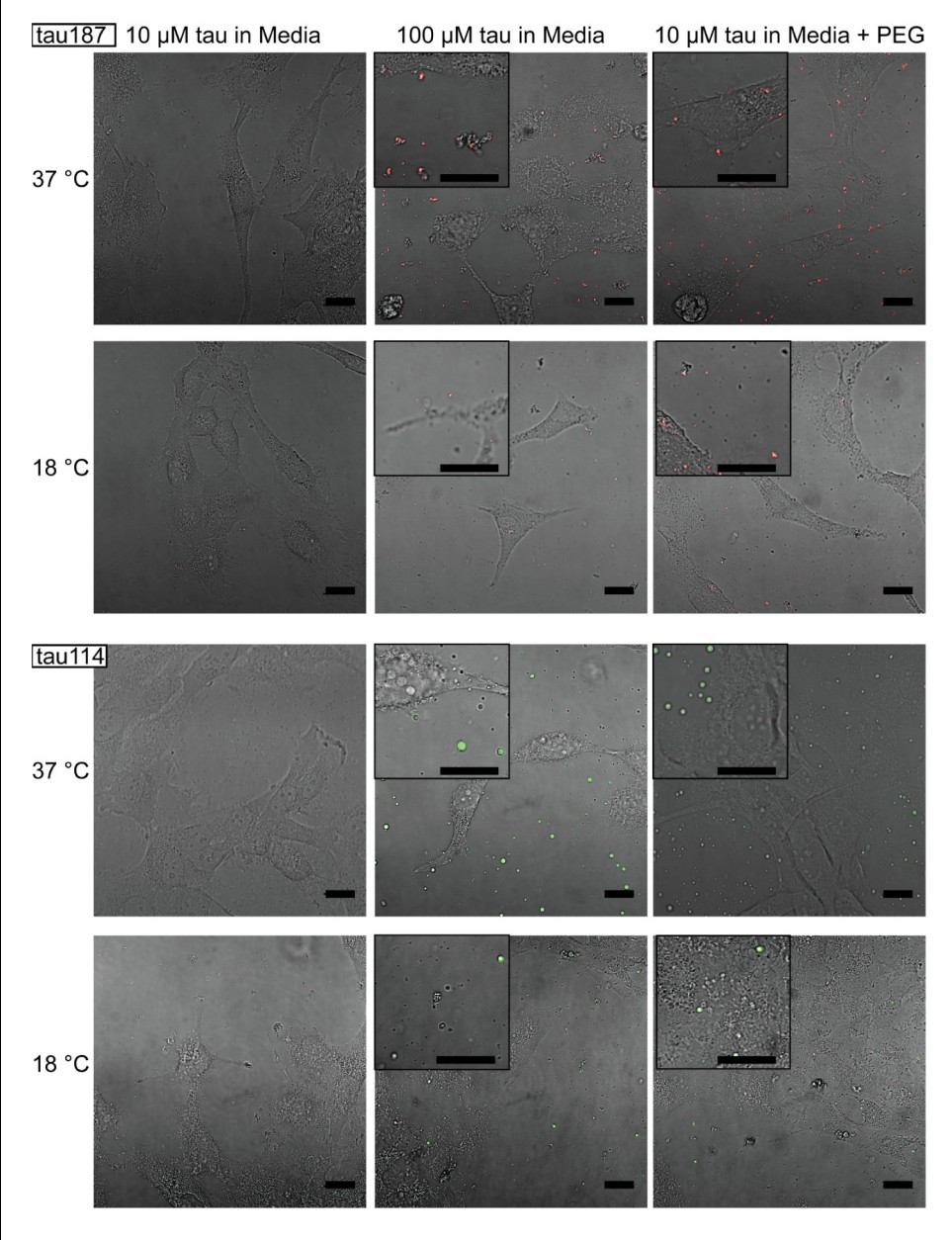

**Figure 7.** Tuning experimental conditions to catch tau-RNA complex coacervates in the presence of living cells. Bright field images and superimposing fluorescence images of tau-RNA CC coculturing with H4 cells, with 10 μM tau (left), 100 μM tau (middle) and 10 μM tau with 10% v.v. PEG (right). Samples at 37 °C (first row) and 18 °C (second row) were images with representative images showing the co-presence of living cells and tau-RNA CCs. Tau187 (Top) and tau114 (Bottom) were used showing tau114 with higher propensity at CC formation. Alexa Fluor 488 was used to prepare fluorescent labeled tau. 3 μg/mL polyU RNA per 1 μM tau was used to prepared samples. Scale bar is 20 μm long.

DOI: https://doi.org/10.7554/eLife.42571.024

theory and simulation to describe the parameter space for LLPS. In what follows, we discuss the relevance of our findings in the context of the physical mechanism of LLPS in vivo.

Although the FH-VO model cannot model spatially varying charges along the peptide backbone, we were able to fit the experimental data by treating the Flory-Huggins χ parameter as an empirical, temperature-dependent, adjustable parameter. This result highlights the fact that the FH-VO model is adaptable to experimental data. Still, the FH-VO model has limited predictability and should be

seen as a qualitative descriptor of phase separation. In contrast, FTS is an approximation-free analysis that can provide physical insight and predictive information for biopolymers, such as scaling relationships and polymer or protein sequence effects. As shown above, the tau-RNA phase diagram was successfully reproduced by FTS using model parameters that are reasonable estimates of the experimental physical conditions. With reasonable estimates for the parameters in our polymer model ($\epsilon_r = 80$, $b = 4$ )), our simulations predict that the lower phase boundary falls in the vicinity of physiological conditions. This finding suggests that FTS can be a powerful theoretical modeling technique to describe and rationalize tau-RNA CC as a competition between short-ranged excluded volume interactions and long-ranged electrostatic interactions.

Consider that we can partition the driving forces of CC as

$$\Delta G^{CC} = \underbrace{\Delta H^{tau/RNA}}_{(-)} - \underbrace{T\Delta S^{comb}}_{(-)} + \underbrace{\Delta H^{ex} - T\Delta S^{noncomb}}_{excluded\,volume\,or\,\chi}$$

where the first two terms are the negative (favorable) enthalpic contribution from tau/RNA interactions and the ideal entropy of mixing term (which is negative because we are considering CC formation). These first two terms are approximately accounted for in the original VO model, and by themselves predict UCST behavior (see SI). The last two terms introduce an non-ionic excess enthalpic contribution and a nonideal, noncombinatoric entropy that are introduced into the FH-VO model through the Flory-Huggins χ parameter, or within FTS through the temperature-dependent excluded volume. Given the experimental observation of LCST phase behavior, these terms must be important and we now estimate their value from our model.

Modeling the LCST experimental tau-RNA CC phase diagram using the FH-VO model by invoking an entropic term in the Flory-Huggins χ parameter, or by FTS using a temperature dependent excluded volume, both provide an estimate of the entropic contribution that drives CC formation. The temperature-dependent excluded volume $v$ used to describe LCST phase behavior within FTS can be formally related to the Flory-Huggins $\chi$ parameter to second order in the polymer volume fractions $v = b^3(1 - 2\chi)$ (**Gennes, 1979**). Substituting our empirical excluded volume, we obtain from FTS an interaction parameter $\chi$ of the form $\chi = \epsilon_s + \epsilon_H/T$, with $\epsilon_s$ being a non-combinatoric entropic term and $\epsilon_H$ an enthalpic term. Introducing conventional units (see Materials and methods for details) gives an unfavorable non-electrostatic enthalpy of phase separation of $\Delta H^{ex}$ = 0.23 kJ • mol$^{-1}$ of monomer, and a favorable noncombinatoric entropy of phase separation of T$\Delta S^{noncomb}$ = 1.1 kJ • mol$^{-1}$ of monomer at T = 300 K. For comparison, the empirical χ from fitting the experimental data with the FH-VO model gives $\Delta H^{ex}$ = 2.3 kJ • mol$^{-1}$ of monomer and T$\Delta S^{noncomb}$ = 3.24 kJ • mol$^{-1}$ of monomer.

Notably, $\Delta H^{ex}$ is small and positive. We hypothesize that the positive, that is nonionic, enthalpy value for forming a coacervate phase is due to the requirement of breaking favorable interactions between hydrophilic residues and water that stabilizes the solution phase of tau ($\Delta H^{ex}$ = -$\Delta H^{tau/water}$). For comparison, the enthalpy of forming a hydrogen bond $\Delta H_{HB}$ at room temperature is ∼ −8 kJ • mol$^{-1}$ (**Silverstein et al., 2000**) while the enthalpy of hydration for a polar amino acid $\Delta H_{hyd}$ is ∼ −60 kJ • mol$^{-1}$; (**Makhatadze and Privalov, 1993**; **Matubayasi, 2017**). Given that $\Delta H^{tau/RNA}$ for tau–RNA association is negative and tau remains hydrated in the CC state (i.e. tau-water interface is not dehydrated), there has to be a source of penalty in the form of a positive $\Delta H^{ex}$ value; the unfavorable $\Delta H^{ex}$ associated with tau-RNA CC might come from the loss of hydrogen bonds in the hydration shell from overlapping and sharing of the tau hydration shells in the dense CC phase.

The T$\Delta S^{noncomb}$ value is also small, positive and of comparable magnitude as $\Delta H^{ex}$, making temperature increase a facile modulator favoring tau-RNA CC. Given the positive value of $\Delta H^{ex}$ for tau-RNA CC, the entropy gain upon phase separation is contributing to the driving force of tau-RNA CC formation (besides the electrostatic correlation energy between the polycationic and polyanionic polymer segments that is the major driving force). Looking to potential origins for positive T$\Delta S^{noncomb}$, we consider the entropy gain of breaking a hydrogen bond of T$\Delta S_{HB}$ ∼ 6 kJ • mol$^{-1}$ (**Silverstein et al., 2000**) and the entropy gain associated with the release of a single water molecule from a hydrated surface of ∼7.5 kJ • mol$^{-1}$ (**Thirumalai et al., 2012**). Given that our FTS study only considered excess ions, but no counterions, while fully capturing the LCST behavior through the excluded volume, $v$, our results are consistent with the hypothesis that competing hydrophilic/hydrophobic interactions are responsible for the LCST behavior (**Feil et al., 1993**;

*Choi and Yethiraj, 2015*; *Martin and Mittag, 2018*). At low temperatures, the attractive interaction between water and hydrophilic residues of the biopolymer stabilize the homogenous phase, but above a critical temperature hydrophobic interactions become dominant, in that it becomes more favorable for water to be released from the polymer surface and hydration shell, and for tau and RNA to associate. In this scenario, the entropy gain comes from the release of bound water into the bulk (private communication with Danielsen SPO, McCarty J, Shea J-E, Delaney KT, Fredrickson GH on the "Small ion effects on Self-Coacervation phenomena in block polyampholytes") due to over-lapping of the hydration shell of tau upon CC. In the literature, the entropy gain of counter ion release (*Dobrynin and Rubinstein, 2001*; *Gummel et al., 2007*; *Muthukumar, 2004*; *Hone et al., 2000*) or compressibility effects (*Lacombe and Sanchez, 1976*; *Sanchez and Lacombe, 1978*) have been proposed as origins for the LCST behavior, and as prevalent driving forces for CC (*Chang et al., 2017*). While this study cannot entirely delineate between these possible contributions that are all subsumed into the Flory-Huggins χ parameter or the excluded volume parameter in FTS, we demonstrate that it is not necessary to invoke a specific mechanism, such as counter ion release—the most popular hypothesis, to rationalize LCST driven CC formation. In fact, we performed FTS studies with (and without) explicit excess ions (*Figure 6C*) observing LCST behavior simply by means of excluded volume and electrostatic considerations and not invoking any counter ion release mechanism to capture the phase diagram of the entropy driven tau-RNA CC. Instead, many factors that globally modulate the excluded volume effects in the biological system of interest and that inevitably modulate the hydration water population, including the hydrophobic effect and crowding, may be considered.

We demonstrated here that tau-RNA CC can be modeled as a coarse-grained polyelectrolyte mixture using equilibrium theory, and revealed the associated driving factors and the different thermodynamic contributions to the phase diagram. However, this finding does not contradict the possibility that tau-RNA complex coacervation is followed by, or even can facilitate, amyloid fibrillization of tau. Comparing our study to previous reports in the literature (*Ambadipudi et al., 2017*; *Zhang et al., 2017*; *Hernández-Vega et al., 2017*; *Wegmann et al., 2018*; *Eschmann et al., 2017*; *Pavlova et al., 2016*), it is clear that tau in fibrils possess dramatically different properties than tau in CCs. In contrast to fibrils, tau-RNA CCs are reversible and tau remains conformationally dynamic – this is because CCs are formed with a stable tau variant, such as the WT-derived tau studied here. However, once aggregation-promoting factors are introduced, not only can the thermodynamically stable phase of tau-RNA CC be driven out of equilibrium, but the dense CC phase harboring high tau and RNA concentration may also lower the activation barrier for, and thus facilitate, tau aggregation. Still, tau complex coacervation is a distinct state and fibrilization is a distinct process, where the equilibrium of one does not contradict with its kinetic transformation into the other. Recently, the possibility of the transformation of tau CCs into tau fibrils has been demonstrated (*Ambadipudi et al., 2017*). We have independently investigated these questions and find that irreversible transformation can be triggered by doping tau-RNA CC with highly sulfated polysaccharide heparin (*Figure 1—figure supplement 4*). Tau is first driven toward an equilibrium complex coacervate state, from which tau can either re-dissolve into solution state reversibly, or form amyloid fibrils when aggregation driving force is present. However, the mechanism by which the CC state of tau influences the rate of aggregation and/or alters the aggregation propensity of tau is not understood, and will and should be the subject of future studies.

The physiological role of tau-RNA CC as a possible regulatory mechanism or as an intermediate toward fibrilization is an ongoing topic of research. In either case, for tau-RNA CC to be relevant for cellular function LLPS would have to be possible near (certain) physiological conditions. Our in vitro experiments found the tau-RNA CC phase diagram boundary to lie near physiological conditions. This suggests that tau-RNA CC can occur in vivo upon modulation of parameters, such as the local temperature, electrostatic balance, including local pH, and osmotic pressure. We demonstrate that indeed tau-RNA CC can be achieved in co-culture with living cells. While the coexistence of tau and RNA at low (10 µM) polymer concentrations is not sufficient to drive CC in cellular media, the addition of a molecular crowding reagent is, under physiological conditions (*Figure 7*). While in this study crowding has been simulated with PEG, many cellular proteins can act as molecular crowding reagents. This data encourages us to speculate that mechanisms that increase the already high concentrations of free proteins and other macromolecular constituents, *not* participating in CC, beyond the normal level within the cell (estimates of 50–200 mg/mL [*Finka and Goloubinoff, 2013*]) could

be sufficient to promote tau-RNA CC by exerting crowding pressure. Thus, biological mechanisms that increase the concentration of intrinsically disordered and charged proteins and nucleic acids may be potent factors that drive liquid-liquid phase separation in the cellular context. Specifically for the context of this study, high concentrations of tau-RNA are by themselves sufficient to drive CC formation (*Figure 7*). Given that tau is known to bind and localize to microtubules in the axons of neurons, it is not a stretch to envision a scenario where the local concentration of tau would be highly elevated under certain stress conditions, around regions like the axon initial segment. We proposed at these places in neuron, tau-RNA CCs have a higher probability to be observed. However, even though our calculations and experimental data support a model where tau-RNA CC in vivo is possible, whether this actually occurs within the cell depends on many other factors, among them the strength of tau-microtubule binding that compete with tau-RNA CC.

## Conclusion

We report here the first detailed picture of the thermodynamics of tau-RNA complex coacervation. The observation of an LCST phase diagram implies that although electrostatic interactions are key to CC formation, factors that contribute to solvation entropy gain are key to driving liquid-liquid phase separation. We have computed the first approximation-free theoretical phase diagram for tau-RNA complex coacervation from FTS, where we introduced a temperature-dependent excluded volume term. Simulations show a competition between electrostatic strength (parameterized by the salt concentration) and excluded volume (parameterized by the solvent quality). This knowledge can be used to design experiments that perturb this parameter space in vivo, as well as predict or understand biological mechanisms that may be favorable towards liquid-liquid phase separation. As a proof of this concept we have shown that by deliberately changing salt concentration, temperature, and solvent quality (by the addition of PEG), we can make tau-RNA LLPS appear or disappear in cellular medium with *live* cells. Interestingly, we find that without any adjustable parameters our simulations predict that tau-RNA is positioned near the binodal phase boundary around physiological conditions. This suggests that small and subtle changes within the cellular environment may be sufficient to induce LLPS in otherwise healthy neurons. Even if the conditions that induce LLPS in the cell is transient, the LLPS state can facilitate irreversible protein aggregation if aggregation-promoting factors are already available, giving credence to the idea that LLPS may play a role in neurodegenerative diseases. However, we speculate that LLPS is reversible in the majority of biological events that drive LLPS, making it hard to observe this state within the cellular context.

# Materials and methods

**Key resources table**

| Reagent type (species) or resource | Designation | Source or reference | Identifiers |
|---|---|---|---|
| Peptide, recombinant protein | tau187 | *Peterson et al., 2008* | |
| Chemical compound, drug | PolyU RNA | Sigma | CAS: 27416-86-0; Cat#: P9528 |
| Chemical compound, drug | MTSL | Toronto Research Chemicals | CAS: 81213-52-7; Cat#: O875000 |
| Cell line (*E. coli*) | *E. coli* BL21 (DE3) | Sigma | Cat#: CMC0014 |

## Protein expression and purification

Unless stated, a 20 mM ammonium acetate buffer at pH 7.0 was used and referred to here as final buffer. Tau, RNA, NaCl, PEG and other stocks were prepared using final buffer. Measurements were taken in final buffer at room temperature unless stated.

N-terminal truncated, microtubule binding domain containing tau187 (residues 255–441 with a His-tag at the N-terminus) were used for in vitro studies. The cloning, expression, and purification have been previously described (*Peterson et al., 2008*; *Pavlova et al., 2009*). The single cysteine variant of tau187 (tau187C291S) were generated via site-direct mutagenesis. *E. coli* BL21 (DE3) cells

previously transfected were cultured from frozen glycerol stock overnight in 10 mL luria broth (LB) which was used to inoculate 1 L of fresh LB. Culturing and inoculation were performed at 37°C with shaking of 200 rpm. At $OD_{600}$ of 0.6–0.8, tau187 variant expression was induced by incubation with 1 mM isopropylß-D-thiogalactoside (Sigma Aldrich) for 2–3 hr. Cells were harvested by centrifugation for 30 min at 5000 × g (Beckman J-10; Beckman Instruments, Inc), and the pellets were stored at −20°C until further use.

Cell pellets were resuspended in lysis buffer (Tris-HCl pH 7.4, 100 mM NaCl, 0.5 mM DTT, 0.1 mM EDTA, 1 mM PMSF) with 1 Pierce protease inhibitor tablet (Thermo Fisher). Lysis was initiated by the addition of lysozyme (2 mg/ml), DNase (20 µg/ml), and $MgCl_2$ (10 mM) and incubated for 30 min on ice. Lysate was then heated to 65°C for 13 min, cooled on ice for 20 min and then centrifuged to remove the precipitate. The supernatant was loaded onto a Ni-NTA agarose column pre-equilibrated with wash buffer A (20 mM sodium phosphate pH 7.0, 500 mM NaCl, 10 mM imidazole, 100 µM EDTA). The column was then washed with 20 ml of buffer A, 15 ml buffer B (20 mM sodium phosphate pH 7.0, 1 M NaCl, 20 mM imidazole, 0.5 mM DTT, 100 µM EDTA). Purified tau187 was eluted with buffer C (20 mM sodium phosphate pH 7.0, 0.5 mM DTT, 100 mM NaCl) supplemented with varying amounts of imidazole increasing from 100 mM to 300 mM. The protein was then concentrated via centrifugal filters (MWCO 10 kDa; Millipore Sigma) and the buffer was exchanged into final buffer by PD-10 desalting column (GE Healthcare). The final protein concentration was determined by UV-Vis absorption at 274 nm using an extinction coefficient of 2.8 $cm^{-1}mM^{-1}$, calculated from absorption of Tyrosine [3].

## Spin labeling and cw EPR

Freshly eluted tau187C291S (with one cysteine at site 322) was replaced in final buffer using a PD-10 desalting column (GE Healthcare). Protein after PD-10 was labeled overnight at 4°C by immediately mixing with a 10-fold molar excess of the spin label (1-oxyl-2,2,5,5-tetramethylpyrroline-3-methyl) methanethiosulfonate (MTSL; Toronto Research Chemicals), resulting in spin labeled tau (tau187C291S-SL). Excess label was removed using PD-10. The protein was concentrated using centrifugal filter (MWCO 10 kDa; Amicon) and the final protein concentration was determined by UV-Vis absorption at 274 nm as mentioned above. Non-labeled tau187C291S was used in order to achieve spin dilution.

Cw EPR measurements were carried out using a X-band spectrometer operating at 9.8 GHz (EMX; Bruker Biospin, Billerica, MA) and a dielectric cavity (ER 4123D; Bruker Biospin, Billerica, MA). 100 µM tau187C291S-SL was mixed with 400 µM tau187C291S to reach 20% spin labeling. Samples under droplet forming condition were prepared by adding 1.5 mg/mL RNA, and tau samples under aggregation-inducing conditions prepared by adding 125 µM heparin (15 kDa average MW; Sigma-Aldrich). A sample of 4.0 µL volume was loaded into a quartz capillary (CV6084; VitroCom) and sealed at both ends with critoseal, and then placed in the dielectric cavity for measurements. Cw EPR spectra were acquired by using 6 mW of microwave power, 0.5 gauss modulation amplitude, 100 gauss sweep width, and 8–64 scans for signal averaging.

## Cw EPR spectra analysis

The recorded cw EPR spectra were subjected to single- or double-component simulation. EPR simulation and fitting were performed using MultiComponent, a program developed by Christian Altenbach (University of California, Los Angeles). For all spectra fitting, the magnetic tensors A and g were fixed and used as constraints as previously reported (*Pavlova et al., 2016*). These values are $A_{xx}$ = 6.2 G, $A_{yy}$ = 5.9 G, $A_{zz}$ = 37.0 G, and $g_{xx}$ = 2.0078, $g_{yy}$ = 2.0058, and $g_{zz}$ = 2.0022.

For soluble tau, the cw EPR spectra were best fitted with a single-component simulation and the rotational diffusion constant (R) can be extracted. The rotation correlation time $\tau_R$ was calculated using $\tau_R$ = 1/(6R). For tau-heparin aggregates, the cw EPR were subjected to double-component simulation, where the parameters of the fitted single-component were used as a mobile-component. The immobile-component were set to be identical to the mobile-component, except the diffusion tensor tilt angle $\beta_D$ = 36° and the order parameter S. The fitting parameters were limited at a minimum, which includes the population, p, rotational diffusion constants of mobile- and immobile-component, $R_1$ and $R_2$, and the order paramter, S of the immobile-component. The fitted immobile-component were used to represent the rotational correlation time for tau-heparin fibrils. For tau-

RNA CC. the cw EPR spectra were subjected to both single- and double-component fitting. Comparing the two fitting schemes showed that singl-component fitting has almost overlapped the cw EPR spectra, while double-component fitting results in a immobile-component population of ~10% (data not shown). This showed that tau-RNA CC cw EPR spectra can be sufficiently fit with single-component. The fitted rotational correlation time was calculated and plotted against tau-heparin samples.

## Turbidimetry and brightfield microscopy

Turbidity of samples at room temperature were represented by optical density at a 500 nm wavelength ($OD_{500}$), using a Shimadzu UV-1601 spectrophotometer (Shimadzu Inc). The amount of coacervates in a sample were approximated to be propotional to its $OD_{500}$.

Tubidity of samples at ramping temperatures were represented by $OD_{500}$ measured using Jasco J-1500 CD Spectrometer (JASCO Inc) equipped with temperature controller and spectrophotometer. 120 μL of 20 μM tau187C291S, 60 μg/mL polyU RNA and 30 mM NaCl in working buffer were prepared in a 100 μL cuvette (Starna Scientific Ltd) and kept at 4°C for 5 min before cycling. Heating and cooling temperatures were ramped at 1°C/min while $OD_{500}$ was monitored.

Bright field images were examed to confirm the presence of tau-RNA CC. 100 μM tau187C291S and 300 μg/mL polyU RNA was mixed in presence of 20 mM ammonium acetate and 30 mM NaCl. 10 μL of the mixture was pipetted onto a microscope slide with a cover slide gapped by two layers of double-sided sticky tape. Temperatures were controlled using an incubator. Bright field images were acquired using a spectral confocal microscope (Olympus Fluoview 1000; Olympus, Center Valley, PA).

## Determining tau-RNA CC composition

It was shown by fluorescence microscopy in protein-RNA LLPS that protein is concentrated inside the droplet (*Patel et al., 2015*; *Elbaum-Garfinkle et al., 2015*). For representing tau inside the droplets with measurement taken from droplet suspension, we quantified the percentage of tau present as droplets. After mixing and centrifuging 60 μL droplet suspension of 400 μM tau187/322C and 1500 μg/mL polyU, ~1 μL dense phase was generated with clear boundary against dilute phase. Dissolving dense phase in high concentration of NaCl resulted in transparent solution thus UV absorption can be measured.

Due to the difficulty of preparing large volume of pure dense phase, we can only underestimate the tau and polyU concentration in dense phase. Since tau and RNA have different UV absorbance spectra, fitting spectra of the tau-RNA mixed sample with those of pure tau and polyU generated the concentration of both. Fitting results showed that over 99% of the tau and over 99.9% of polyU were condensed inside the dense phase. This partitioning guaranteed that the property of tau in the droplet suspension represents those in the droplets.

## Cell culture and confocal microscopy

Protein (tau187 or K18) was labeled with Alexa Fluor 488 or 555 5-SDP ester (Life Technologies) according to the suppliers instructions. After labeling, 100 mM glycine was added to quench the reaction and the proteins were subjected to Zeba desalting columns (Thermo Scientific) to remove any unreacted label. Average label incorporation was between 1 and 1.5 moles/mole of protein, as determined by measuring fluorescence and protein concentration ($A_{max} \times$ MW of protein / [protein]$\times \varepsilon_{dye}$).

H4 neuroglioma cells (ATCC HTB-148) were cultured in DMEM supplemented with 10% FBS, 100 μg/mL penicillin/streptomycin. Cultures were maintained in a humidified atmosphere of 5% $CO_2$ at 37°C. Tau protein (1:20 labeled K18:unlabeled tau114), RNA, PEG, and media (DMEM,10% FBS,1% Pen/Strep) were mixed at the indicated concentrations and added to cells at varying temperatures. Images were obtained on a Leica SP8 Resonant Scanning Confocal.

## Tau in vitro phosphorylation

Phosphorylation of tau was performed as previously described (*Despres et al., 2017*). In brief, tau protein (40 μL at 6 mM) was mixed with 200 μL mouse brain extract and incubated overnight at 37°C in phosphorylation buffer (40 mM HEPES pH 7.3, 2 mM $MgCl_2$, 5 mM EGTA, 2 mM DTT, 2 mM ATP,

1 µM okadaic acid, protease inhibitors). After incubation, samples were centrifuged and the supernatant was buffer exchanged using zeba desalting columns (Thermo Fisher) into buffer (20 mM ammonium acetate, pH 7). Concentration was determined by BCA assay. Phosphorylation was confirmed using a western blot assay to look at phospho-epitopes 396/404 using PHF-1 Antibody (Peter Davies).

## Flory-Huggins-based Voorn-Overbeek (FH-VO) modeling

FH-VO is based on a Flory-Huggins (FH) treatment, where the polymer system is mapped onto a lattice. Voorn and Overbeek extended the FH formalism to polyelectrolytes by including long-ranged electrostatic interactions with a Debye–Hückel term. The resulting expression for the free energy of mixing (ΔGmix) per lattice site is

$$\frac{\Delta G_m}{M k_B T} = \sum \frac{\phi_i}{N_i} ln\phi_i - \alpha[\Sigma \sigma_i \phi_i]^{\frac{3}{2}} + \Sigma \chi_{ij} \phi_i \phi_j \tag{S1}$$

where $M = V/(l_w)^3$ is the total number of lattice sites. In *Equation S1*, the index $i$ refers to one of the five species. $N_i$ is the degree of polymerization for species $i$. For tau187 and tau114, $N_i$ equals to the length of the polypeptides (*Figure 1—source data 1*); while for RNA, $N_i$ is estimated by the average MW 900 kDa for polyU RNA and the MW of condensated uridine monophosphate, 306 Da. For monovalent ions and water, $N_i = 1$.

$\sigma_i$ is the average charge per monomer, which is determined by (net charge)/$N_i$. The net charges of tau at experimental pH conditions (pH = 7) were estimated based on primary sequences in *Figure 1—source data 1*, using pepcalc.com. $\sigma_i$ for other species were listed in *Figure 2—source data 1*. In FH-VO model, $\sigma_i$ is fixed. We also consider a modified version, a FH-VO-CR model, where $\sigma_i$ of RNA is set to a function of temperature as discussed further below.

In *Equation S1*, $\phi_i$ is the volume fraction of species (tau, RNA, Na$^+$, Cl$^-$, H$_2$O). $\phi_i$ was computed by $\phi_i = c_i \times N_i \times \frac{1}{c_w}$ where $c_i$ is the molar concentration and $c_w$ the molar concentration of pure water computed from water volume molarity: $c_w$ = 55.56 mol/L. In experiments, $c_{tau}$ and $c_{RNA}$ were designed to reach a 1:1 charge ratio, therefore, we have $N_{RNA} \times c_{RNA} = 11 \times c_{tau} = 11 \times [tau]$. In addition to NaCl, there is 20 mM ammonium acetate in the buffer. The total monovalent salt concentration is $c_{salt} = c_{NaCl} + 20\,mM = [NaCl] + 20\,mM$. Therefore, $\phi_i$ were calculated from experimental [tau] and [NaCl] as,

$$
\begin{aligned}
\phi_{tau} &= [tau] \times 207 \times \frac{1}{c_w} \\
\phi_{RNA} &= [tau] \times 11 \times \frac{1}{c_w} \\
\phi_{salt} &= ([NaCl] + 20\,mM) \times \frac{1}{c_w} \\
\phi_{polymer} &= \phi_{tau} + \phi_{RNA} = [tau] \times 218 \times \frac{1}{c_w} \\
\phi_{water} &= 1 - \phi_{polymer} - \phi_{salt}
\end{aligned}
\tag{S2}
$$

$\alpha$ is the strength of the electrostatic interactions defined as

$$\alpha = \frac{2}{3}\sqrt{\frac{\pi}{l_w^3}} \left(\frac{e^2}{4\pi \epsilon_r \epsilon_0 k_B T}\right)^{3/2} \tag{S3}$$

where $l_w$ is the length of a lattice, computed from $c_w$, $l_w = \sqrt[3]{\frac{1 \times 10^{-3} m^3}{c_w N_A}}$, $\epsilon_r \epsilon_0$ the water permitivity, $\epsilon_r \epsilon_0 = 80 \times 8.85 \times 10^{-12}\,F/m$, $k_B$ the Boltzmann constant and T the absolute temperature.

$\chi_{ij}$ is the Flory-Huggins interaction parameter between species i and j, which will be defined and discussed below.

The three terms on the right-hand side of *Equation S1* are respectively: (1) the ideal Flory-Huggins mixing entropy, (2) the mixing enthalpy due to Coulombic interactions based on Debye–Hückel approximation (*Hückel and Debye, 1923*) and (3) the excess free energy to account for the non-Coulombic interactions, which can include contributions from water perturbation (*Fu and Schlenoff, 2016*), cation-π interaction (*Kim et al., 2016*) and dipole-dipole interactions (*Holehouse et al., 2015*). *Equation S1* has been successfully applied in PDMAEMA-PAA complex coacervate

(*Spruijt et al., 2010*). In this work, we refer to *Equation S1* as FH-VO model, which is a minimal model for complex coacervation.

## Determining phase separation temperature

A phase separation temperature, $T_{cp}$, was assigned to the cloud point of the sample. $T_{cp}$ was determined by fitting normalized turbidity-temperature curves to a sigmoid function as follows

$$normalized\ turbidity = \frac{1}{1 + exp\big(-k \times \big(T - T_{cp}\big)\big)},$$

to find

$$T = T_{cp}$$

## FH-VO binodal curve computation

$\phi_i$ and T can be converted from/to experimental conditions as described, where tau and RNA are added at a fixed charge neutrality ratio. Therefore, $\Delta G_{mix}$ depends on four variables: total polymer volume fraction $\phi_{polymer} = \phi_p + \phi_q$, total salt volume fraction $\phi_{salt} = \phi_{s+} + \phi_{s-}$, temperature $T$ and X, a matrix of $\chi_{pp},\ \chi_{pq},\ \chi_{ps+}, \ldots, \ \chi_{qp},\ \chi_{qq},\ \ldots$. A two-phase equilibrium exists where the sum of mixing free energy of two coexisting phases are lower than that of the homogeneous mixture. For simplicity, we adopt the assumption that the salt concentration in both two phases are identical (*Spruijt et al., 2010*), leaving the system a binary mixture of polymer and buffer. Binodal compositions are defined by pairs of points on the curve of $\Delta G_{mixing}$ vs. $\phi_{polymer}$ that have common tangents, corresponding to compositions of equal chemical potentials of both buffer and polymer in dense and dilute phases.

A binodal composition curve (binodal curve) was computed by finding the bi-tangent points of $\Delta G_{mixing}$ vs. $\phi_{polymer}$ at a series of $\phi_{salt}$ at given temperature $T$ and given parameters. Given $\phi_{salt}, T$ and X, the mixing free energy is solely dependent on $\phi = \phi_{polymer}$:

$$f(\phi) = \Delta G_{mixing}\big(\phi_{polymer}\big)$$

A bi-tangent pair $(\phi_1, f_1)$, $(\phi_2, f_2)$ was calculated by solving the set of nonlinear equations (*Rubinstein and Colby, 2003*; *Kwon et al., 2015*),

$$\begin{cases} \frac{\partial}{\partial \phi}|_{\Phi = \Phi_1} & -\frac{\partial}{\partial \phi}|_{\Phi = \Phi_2} = 0 \\ \big(f - \Phi \frac{\partial}{\partial \phi}\big)|_{\Phi = \Phi_1} & -\big(f - \Phi \frac{\partial}{\partial \phi}\big)|_{\Phi = \Phi_2} = 0 \end{cases}$$

which was solved by R function *nleqslv* using Newton-Ralphson algorithm at given initial guess. Finally, the $\phi_{polymer}$ and $\phi_{salt}$ were converted into [tau] and [NaCl] as described.

## Coarse-grained polyelectrolyte model used in FTS

Our system consists of $n$ total polymers made up of $n_\tau$ tau molecules of length $N_\tau$ and $n_p$ RNA molecules of length $N_p$. Each amino acid is treated as a single Kuhn segment of length $b$. The solvent is treated implicitly with a uniform dielectric background $\epsilon$. For simplicity, we only consider the symmetric case of $N_p = N_\tau$. Chain connectivity is enforced by a harmonic bond potential of the form $\beta U_{bond} = \frac{3}{2b^2} \sum_{\alpha=1}^{n} \sum_{j=1}^{N} \big(|\boldsymbol{r}_{\alpha,j} - \boldsymbol{r}_{\alpha,j-1}|\big)^2$ where $\boldsymbol{r}_{\alpha,j}$ is the coordinates of bead $j$ on chain $\alpha$. In addition to chain connectivity, all monomers interact with a short-ranged excluded volume potential (*Doi and Edwards, 1988*). We take the well-known Edward's delta function model for the excluded volume interaction $\beta U_{ex} = v\delta(\boldsymbol{r})$ where $v$ is the excluded volume parameter (*Doi and Edwards, 1988*). The charge of each bead $j$ for the tau molecule $z_{\tau,j}$ is determined from the primary amino acid sequence with aspartic (D) and glutamic (E) acid being $z_{\tau,j} = -1$, arginine (R) and lysine (K) being $z_{\tau,j} = +1$ and all other amino acids being neutral $z_{\tau,j} = 0$. The RNA chain is treated as a fully-charged chain with $z_{p,j} = -1$ for all monomers. Charged segments interact via a long-ranged Coulomb potential $\beta U_{el} = \frac{l_B z_i z_j}{r}$ with $l_B = \frac{e^2}{4\pi\epsilon_0\epsilon_r k_B T}$ being the Bjerrum length, $e$ is the unit of electronic charge, $\epsilon_0$ is the

vacuum permittivity, and $\epsilon_r$ is the dielectric constant. For a schematic depiction of the polymer physics model see *Figure 3* in the main text.

The model is 'regularized' by smearing all statistical segments over a finite volume instead of treating them as point particles (*Delaney and Fredrickson, 2016*). This is accomplished by endowing each bead with a Gaussian profile with a width on the order of the statistical segment length $\Gamma(r) = (3/\pi b^2)^2 \exp(-3r^2/b^2)$ where $r$ is a radial distance from the monomer center. As a consequence of this density smearing, the interactions between monomers 'softens' (*Villet, 2012*).

## Transformation of particle model to a statistical field theory

The advantage of the coarse-grained polyelectrolyte model employed in this work is that it can be exactly converted to a statistical field theory by utilizing a Hubbard-Stratonovich transformation as described in *Fredrickson (2006)*. Invoking this transformation, the canonical partition function is expressed in terms of two fluctuating auxiliary fields $w$ and $\varphi$ which serve to decouple the excluded volume and Coulombic interactions, respectively (*Patel et al., 2015*; *Li et al., 2018b*; *Kwon et al., 2014*; *Lee et al., 2016*; *Boeynaems et al., 2017*). In the statistical field representations the canonical partition function is

$$Z = Z_0 \int Dw \int D\varphi \exp(-H[w,\varphi]) \tag{S4}$$

where $Z_0$ contains the ideal gas partition function and self-interaction terms. The field-theoretic Hamiltonian for this model is

$$H[w,\varphi] = \frac{1}{2v}\int d\boldsymbol{r}\, w(\boldsymbol{r})^2 + \frac{1}{8\pi l_B}\int d\boldsymbol{r}\, |\nabla\varphi|^2 - n_\tau \ln Q_\tau[w,\varphi] - n_p \ln Q_p[w,\varphi] \tag{S5}$$

where $Q_\tau[w,\varphi]$ and $Q_p[w,\varphi]$ are the partition functions for a single tau and a single RNA molecule in the conjugate fields. These single chain partition functions can be computed using a Gaussian chain propagator such that

$$Q_l[\psi] = \frac{1}{V}\int d\boldsymbol{r}\, q_l(\boldsymbol{r}, N_l; \psi) \tag{S6}$$

where $l$ indexes the chain type (tau/RNA) and $\psi(j) = i\,\Gamma_\star(w + z_j\varphi)$ with $i = \sqrt{-1}$ and $\star$ a spatial convolution. The chain propagator $q_l(\boldsymbol{r}, j; \psi)$ is constructed from a Chapman-Kolmogorov-type equation

$$q_l(\boldsymbol{r}, j+1; \psi) = \left(\frac{3}{2\pi b^2}\right)^{3/2} \exp[-\psi(\boldsymbol{r}, j+1)] \int d\boldsymbol{r}'\, q_l\left(\boldsymbol{r}', j; \psi\right) \exp\left(-\frac{3|\boldsymbol{r}-\boldsymbol{r}'|^2}{2b^2}\right) \tag{S7}$$

with initial condition $q_l(\boldsymbol{r}, 0; \psi) = \exp[-\psi(\boldsymbol{r}, 0)]$. From the field theoretic Hamiltonian any thermodynamic observable may be computed as an ensemble average of a corresponding operator expressed in terms of the field configurations $\tilde{G}[w,\varphi]$

$$\langle G \rangle = \frac{Z_0}{Z}\int Dw \int D\varphi\, \tilde{G}[w,\varphi] \exp(-H[w,\varphi]) \tag{S8}$$

We stress that no additional approximations are made in moving from a particle-based model to a statistical field theory. The advantage of such a transformation is that the pairwise interactions between monomers are decoupled in favor of interactions between monomers and a complex-valued field. This transformation is particularly suited to our purposes here as conventional particle simulations can only study the earliest stages of protein aggregation.

## Field theoretic simulations using CL sampling

Field theoretic simulation (FTS) has been widely used to model synthetic polymers (*Delaney and Fredrickson, 2016*), including LLPS in polyelectrolytes and polyampholytes (*Delaney and Fredrickson, 2017*; *Lee et al., 2008*; *Popov et al., 2007*). The interested reader is directed to this literature for further detail of the method. Here we apply this powerful numerical method in a new context to model LLPS of a biological system (tau-RNA under cellular conditions). The main advantage of this

approach is that it does not rely on analytical approximations, and thus can be useful when comparing directly to experiment at conditions where such approximate theories break down. FTS allows one to numerically compute ensemble averages of the form of *Equation S8* by sampling along a stationary stochastic trajectory in the space of the field variables. The method has been presented in detail elsewhere (*Delaney and Fredrickson, 2016*; *Fredrickson, 2006*; *Alexander-Katz et al.,* *2005*). We use complex Langevin (CL) sampling (*Klauder, 1983*; *Parisi, 1983*) to stochastically sample the auxiliary fields. The method involves promoting the fields to be complex-valued and numerically propagating the CL equations of motion

$$\frac{\partial w(\boldsymbol{r},t)}{\partial t} = -\lambda_w \frac{\delta H[w,\varphi]}{\delta w(\boldsymbol{r},t)} + \eta_w(\boldsymbol{r},t)$$

$$\frac{\partial \varphi(\boldsymbol{r},t)}{\partial t} = -\lambda_\varphi \frac{\delta H[w,\varphi]}{\delta \varphi(\boldsymbol{r},t)} + \eta_\varphi(\boldsymbol{r},t)$$

(S9)

where $\eta_w(\boldsymbol{r},t)$ and $\eta_\varphi(\boldsymbol{r},t)$ are real-valued Gaussian white-noise random variables with zero mean and variance proportional to the dissipative coefficients $\lambda_w$ and $\lambda_\varphi$. A single FTS step involves computing the single chain partition functions for a given field configurations $(w,\varphi)$ given by Equation S6 along with any operators $\tilde{G}[w,\varphi]$ followed by updating the field configurations according to Equation S9. Under the condition that the system is ergotic, ensemble averages are computed as time averages over the CL trajectory. All FTS-CL simulations were performed in reduced units by scaling spatial lengths by a reference distance $R_0 = b/\sqrt{6}$ corresponding to the prefactor in the scaling relation of an ideal homopolymer radius of gyration with respect to the chain length $R_g = R_0 N^{1/2}$ (*Flory and Volkenstein, 1969*). Simulations were performed in a cubic box of length $L = 34.0\,R_0$ using periodic boundary conditions. Fields were sampled with a spatial collocation mesh of 32 (*Anderson and* *Kedersha, 2006*) sites. An exponential time difference (ETD) algorithm (*Villet and Fredrickson,* *2014*; *Düchs et al., 2014*) with $\Delta t = 0.01$ was used to numerically propagate the CL equations of motion Equation S13. All simulations were performed on NVIDIA Tesla M2070 or K80 graphics processing units (GPUs).

The thermodynamic state of the system is fully determined by specifying a dimensionless excluded volume parameter $B = v/R_0^3$, a dimensionless Bjerrum length $E = 4\pi l_B/R_0$, and a dimensionless polymer chain number density $C = \rho R_0^3$ with $\rho = \sum_l n_l N_l / V$ where $l$ indexes the chain type (tau or RNA). Additionally, we require the fraction of chains of each type $\phi_l = n_l N_l / \sum_l n_l N_l$. In this work, we consider only a 1:1 charge ratio, which for the model shown in Figure 1 corresponds to $\phi_\tau = 0.954$ and $\phi_p = 0.046$.

## Determination of phase equilibria from FTS

In order to compute the phase coexistence points from FTS, we need to compute the chemical potential $\mu$ and the osmotic pressure $\Pi$. The chemical potential operator for chain type $l$ is $\tilde{\mu}_l = \ln \frac{\rho R_0^3 \phi_l}{N_l} - \ln Q_l$. For the pressure operator, we use the form given in Appendix B of *Delaney and* *Fredrickson (2017)*. The conditions for the stable coexistence of two phases is given by the chemical equilibrium condition $\sum_l \nu_l \mu_l^I = \sum_l \nu_l \mu_l^{II}$ and the mechanical equilibrium condition $\Pi^I = \Pi^{II}$. The stoichiometric coefficient $\nu_l$ ensures charge neutrality. The procedure we employ in this work is that of *Delaney and Fredrickson (2017)* and involves computing the chemical potential and pressure for a range of polymer concentrations. Figure 5-figure supplement 1 (Left) shows a plot of the osmotic pressure vs. the chemical potential for different polymer conentrations. The simulation data represent three branches: a dilute branch (red), a conentrated branch (blue), and an unstable branch (orange). The equilibrium condition of equal chemical potential and equal osmotic pressure can be directly gleaned from the intersection of the dilute and concentrated branch. This gives the critical conditions for phase coexistence. *Figure 5—figure supplement 1* (Right) show a plot of the chemical potential vs. polymer concentration. The critical chemical potential value is shown by the dashed horizontal line. The intersection of this line with the polymer concentration data points from FTS gives the dilute supernatant polymer concetration $\rho^I$ and the coacervate concentration of the

coexisting phase $\rho^{II}$. By repeating this procedure for many different thermodynamic conditions, we can construct the phase diagrams shown in *Figure 5* of the main text.

## Calculation of non-ionic entropy and enthalpy of coacervation

As discussed in the main text, the interaction parameter is decomposed into an entropic and enthalpic contribution $\chi = \epsilon_s + \epsilon_H/T$. According to the Flory-Huggins treatment, the non-combinatoric contribution to the Gibbs free energy of mixing is

$$\Delta G_{mix} = \mathrm{RTn}_p \chi \phi_w$$

where R is the ideal gas constant, T is the temperature, $\mathrm{n}_p$ is the total number of moles of monomer units, and $\phi_w$ is the volume fraction of water. From the relation $\Delta S_{mix} = -\frac{\partial \Delta G_{mix}}{\partial T}$, the non-ideal entropy of mixing is

$$\Delta S_{mix} = -\mathrm{Rn}_p \phi_w \epsilon_s.$$

From the relation $\Delta H_{mix} = \Delta G_{mix} + T\Delta S_{mix}$, the enthalpy of mixing arising from non-ionic interactions is

$$\Delta H_{mix} = \mathrm{Rn}_p \phi_w \epsilon_H.$$

Values in the main text are computed using a water volume fraction of $\phi_w = 0.722$. See *Figure 2— source data 2* for further details.

## Acknowledgements

We acknowledge insightful and sustained discussions with Dr. Xumei Zhang on LLPS of tau under cellular conditions. JM thanks Scott PO. Danielsen for helpful discussions regarding FTS. We acknowledge the use of the NRI-MCDB Microscopy Facility at UC, Santa Barbara. KSK, SH, JM and YL acknowledge support for this work by the National Institute on Aging (NIA) of the National Institute of Health (NIH) through Grant # RO1AG05605. KSK and SH also acknowledge partial support by the Tau consortium of the Rainwater foundation. JES acknowledges partial support by the National Science Foundation (NSF) Grant # MCB-1716956. GHF, KD, JES and JM acknowledge partial support by the MRSEC program of the NSF through Grant # DMR-1720256 (IRG-3). The research reported here made use of support from the Center for Scientific Computing from the CNSI, MRL: an NSF MRSEC (DMR-1720256) and NSF CNS-1725797.

## Additional information

### Funding

| Funder | Grant reference number | Author |
| --- | --- | --- |
| National Institute on Aging | RO1AG05605 | Yanxian Lin<br>James McCarty<br>Kenneth S Kosik<br>Songi Han |
| National Science Foundation | MRSEC Program, Award No. DMR 1720256 | James McCarty<br>Kris T Delaney<br>Glenn H Fredrickson<br>Joan-Emma Shea |
| Santa Barbara Foundation | Tri-Counties Blood Bank Fellowship | Jennifer N Rauch |
| Rainwater Foundation | Tau Consortium | Kenneth S Kosik<br>Songi Han |
| National Science Foundation | MCB-1716956 | Joan-Emma Shea |

The funders had no role in study design, data collection and interpretation, or the decision to submit the work for publication.

## Author contributions
Yanxian Lin, James McCarty, Conceptualization, Data curation, Software, Formal analysis, Validation, Investigation, Visualization, Methodology, Writing—original draft, Writing—review and editing; Jennifer N Rauch, Validation, Investigation, Writing—review and editing; Kris T Delaney, Resources; Kenneth S Kosik, Conceptualization, Supervision, Funding acquisition, Writing—review and editing; Glenn H Fredrickson, Methodology, Writing—review and editing; Joan-Emma Shea, Conceptualization, Formal analysis, Supervision, Funding acquisition, Methodology, Writing—original draft, Writing—review and editing; Songi Han, Conceptualization, Formal analysis, Supervision, Funding acquisition, Writing—original draft, Project administration, Writing—review and editing

## Author ORCIDs
Yanxian Lin (iD) http://orcid.org/0000-0002-9902-1885
James McCarty (iD) http://orcid.org/0000-0002-3838-6004
Songi Han (iD) http://orcid.org/0000-0001-6489-6246

## Decision letter and Author response
Decision letter https://doi.org/10.7554/eLife.42571.027
Author response https://doi.org/10.7554/eLife.42571.028

# Additional files

## Supplementary files
• Transparent reporting form
DOI: https://doi.org/10.7554/eLife.42571.025

## Data availability
All data generated or analysed during this study are included in the manuscript and supporting files.

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
