## [Decision Letter]

Thank you for submitting your article "Narrow equilibrium window for complex coacervation of tau and RNA under cellular conditions" for consideration by *eLife*. Your article has been reviewed by three peer reviewers, one of whom is a member of our Board of Reviewing Editors, and the evaluation has been overseen by Naama Barkai as the Senior Editor. The following individual involved in the review of your submission has agreed to reveal his identity: Carlos Castaneda.

The reviewers have discussed the reviews with one another and the Reviewing Editor has drafted this decision to help you prepare a revised submission.

Summary:

This is an interesting and thorough study that combines experiment, theory, and simulations to understand the physical basis of complex coacervation of tau-protein and RNA. This work established a phase diagram for the tau:RNA polymer system experimentally using cloud point temperatures as a function of polymer (tau:RNA) and salt concentrations and modeled the coacervation of this system to match the experimental data using coarse-grain field theoretic simulations (FTS). This work provides a new framework to understand the thermodynamic underpinnings of complex coacervation for the tau:RNA system.

Essential revisions:

1) Important to this study was the collection of "equilibrium" phase transition experiments for cloud point temperature measurements. A more accurate determination of equilibrium perhaps would be to sit at a particular temperature and obtain the turbidity values rather than perform a temperature ramp experiment where temperature was increased 0.5C/min or 1C/min. This is an issue that concerns the experimental reproducibility (such as the number of experiments collected at each protein concentration/salt concentration). For example, in Figure 2B, it appears that there were 5 independent experiments at 30 mM NaCl and 5 μm Tau w/ RNA, that generated T_cp_ values between 14 and 20 °C? The factors that could give rise to this variability need to be discussed. This is important for these data to be used to test computational models. One concern is the use of ammonium acetate as a buffer at pH 7 for all of their experimental data. The understanding is that ammonium acetate is used to buffer at low pH (acetic acid/acetate pKa 4.7) and high pH (ammonium pKa 9.2), but not at pH 7. How do the authors control for pH effects here? Could the pH effect be a factor that underlies the apparent discrepancy in Figure 2B?

2) Subsection “Tau-RNA complex coacervate phase diagram”, last paragraph: The experimental results indicate that tau114 (134 residues, Figure 1—source data 1) has a higher propensity to undergo liquid-liquid phase separation (LLPS) with RNA than tau187 (207 residues). Can the authors suggest a plausible physical reason for this behavior? If one considers homopolymer LLPS or simple coacervation, propensity to LLPS tends to decrease with decreasing chain length, which is not being observed here for tau-RNA. Are there differences in sequence-specific features between tau114 and tau187 that may provide a plausible rationalization for the experimentally observed trend?

3) In the authors' treatment using Flory-Huggin-Voorn-Overbeek (FH-VO, subsection “Flory-Huggins-Voorn-Overbeek Fit to Experimental Phase Diagram”), a temperature-dependent Flory chi parameter (Equation 1, see the aforementioned subsection) is used to fit the data and to produce the experimental observation that the tau-RNA LLPS tendency increases with increasing temperature (i.e., the system has a LCST). This phenomenon, which is also observed for elastin, is similar to cold denaturation of globular proteins, and its possible physical origin has been discussed recently in the context of biomolecular LLPS by Lin et al., 2018; see Figure 2C of this reference] in terms of the expected temperature-dependence of hydrophobic interactions. The strength of effective attraction between nonpolar groups tends to increase with increasing temperature, as has been demonstrated from model compound transfer experiments, a fact that has long been applied to theory of protein conformational transitions, see, e.g., Dill, Alonso and Hutchinson, 1989. The same trend has also been seen in explicit-water simulations of potentials of mean force of nonpolar solutes in water, see, e.g., Figure 4 in Dias and Chan, 2014. Beyond introducing the temperature-dependent chi parameter as a mere necessity for fitting data, it would be very helpful to the readers to include this perspective (and the above-cited references) in the authors' Discussion to provide a plausible physical rationalization for the observed trend.

4) The statement "high crowding reagents (acting to lower the excluded volume)" is problematic. What do the authors mean by the statement? Could the current formulation of this statement be related to the aforementioned somewhat confused concept about excluded volume presented in the manuscript? Would it make more sense to say that crowding agents act to lower the chain configuration entropy of the dilute phase?

5) In the field theoretic treatment (subsection “Field theoretic simulations of a coarse-grained model of tau-RNA complex coacervation”), the temperature dependent attraction in the form of (1-theta/T) is presented as a temperature-dependent excluded-volume potential. Although it may be mathematically convenient to introduce temperature dependence as a modified "excluded volume" term, doing so is physically inaccurate and conceptually misleading. Excluded-volume interaction is a hard-core repulsion. Inter-chemical-group attractive interactions such as hydrophobic effect do not abolish the hard cores of the atoms. They just affect the potential energy landscape close to but beyond the hard cores. With this in mind, while the usage of a single Gaussian function for this "excluded volume" energy may be a mathematical simplification that is necessary for field theoretic formulations [the Γ(r) function in the last paragraph of the subsection “Coarse grained polyelectrolyte model used in FTS”], it is important to recognize that it is not entirely physically accurate (e.g., one does not use such a functional form in most explicit-chain molecular dynamics force fields). The reason is that in the authors' set up the molecular hard cores can be penetrated but in reality they cannot, and this can have consequences on the predictions of volume fractions, etc. Casting the solvent-mediated attraction as a reduction in excluded volume is also conceptually inconsistent with FH-VO because the FH excluded volume is approximately accounted for by the mixing entropy term, and the introduction of the chi parameter does not envision penetrating any excluded volume. Although this point is largely conceptual in the present context and the reviewers do not suggest redoing the field theory simulation using a more realistic excluded volume potential, it is still importance to have conceptual clarity in the authors' Discussion so as to avoid, e.g., oxymoronic statements such as “the excluded volume becomes attractive”. With a recognition of this limitation, the authors may also wish to qualify their statement regarding the "approximation-free" nature of their simulations (it may be approximation-free with regard to the assumed model potential but the model potential itself is limited at least as described above).

6) Discussion, third paragraph, schematic equation: It would be much clearer to view the overall attractive interactions among tau proteins and RNA as an "effective free energy of interaction" that contains both enthalpic and entropic parts (wherein the entropic contribution comes largely from solvent entropy), and this effective interaction free energy [which, by the way, correspond to the vertical axis in energy landscape drawing for protein folding] is then balanced by configurational/conformational entropy of the chains (which, in the FH-VO approximation is given by the classical mixing term). The authors may wish to adopt such a narrative.

7) More discussions about controls relevant to their FTS simulations is needed to evaluate the FTS methodology.

8) When the authors compared simulation to experiment (subsection “Comparison between simulation and experiment”), how was the Bjerrum length (*l_B_*) parameter obtained from Equation 3 (there is no Equation 3 in the manuscript)? How Equation 2 can be used to obtain *l_B_* if the relative dielectric constant term still needs to be determined? This is a very important section of the paper and a more detailed discussion is needed here.

9) In the conclusion, the authors state that the simulation provides "knowledge that can be used to design experiments that perturb this parameter space in vivo". It would be extremely useful if the authors test FTS and their results by considering tau phosphorylation. Even just modeling tau phosphorylation without RNA would be useful. Additional discussion along this line, if attainable in a timely fashion, is preferred, but not required by the reviewers and the editors.

10) In Figure 6C, for the 120mM NaCl data, the simulated transition temperatures (open green squares) have a significantly higher rate of decrease with respect to increasing [tau] than the corresponding experimental data (blue circles). Could the authors provide a plausible reason for this discrepancy? The Bjerrum length *l_B_* is plotted in Figure 6A. Could the authors confirm in the figure legend that this is just a plot of Equation 2 and no temperature dependence of the relative permittivity *ε_r_* was introduced in Figure 6A?

---

## [Author Response]

Essential revisions:1) Important to this study was the collection of "equilibrium" phase transition experiments for cloud point temperature measurements. A more accurate determination of equilibrium perhaps would be to sit at a particular temperature and obtain the turbidity values rather than perform a temperature ramp experiment where temperature was increased 0.5C/min or 1C/min. This is an issue that concerns the experimental reproducibility (such as the number of experiments collected at each protein concentration/salt concentration). For example, in Figure 2B, it appears that there were 5 independent experiments at 30 mM NaCl and 5 μm Tau w/ RNA, that generated T_cp_ values between 14 and 20 °C? The factors that could give rise to this variability need to be discussed. This is important for these data to be used to test computational models. One concern is the use of ammonium acetate as a buffer at pH 7 for all of their experimental data. The understanding is that ammonium acetate is used to buffer at low pH (acetic acid/acetate pKa 4.7) and high pH (ammonium pKa 9.2), but not at pH 7. How do the authors control for pH effects here? Could the pH effect be a factor that underlies the apparent discrepancy in Figure 2B?

We recognize the reviewer’s concerns that there is a certain level of variability in our data that constitute the phase diagram. Repetition of experimental data points confirms that some level of variability appears inherent to these measurements, and that we are currently unable to control all sources to reduce this variability. We discuss two potential sources underlying this variability.

We observed a steady decrease of turbidity when incubating the tau-RNA mixture under LLPS conditions. We found that the addition of an RNase inhibitor slowed down the decrease in turbidity, while the presence of RNase enzyme accelerated this trend. Based on our control experiments, we conclude that the steady decrease of turbidity results from RNA degradation from RNase contamination. RNase degradation is difficult to control in practice, and can easily contribute to the variability observed, as the reviewer correctly pointed out.

We also took up the reviewer’s comment to avoid ammonium acetate to buffer at pH 7. Since we had used ammonium acetate buffer for our measurements, we tested the pH response of ammonium acetate in the presence of tau or RNA alone. We found in the experimental range of tau and RNA concentration (20 μM tau, 60 μg/mL RNA) that the pH drifted from 6.72 to 6.62 and 6.68 for tau and RNA respectively. Adding the same concentration of tau or RNA shifts HEPES buffer pH from 7.06 to 7.02 (Author response image 1). Comparing ammonium acetate buffer with HEPES buffer, we consider the pH drift of ammonium acetate is measurable, but insignificant. This pH range determines the range of change in the net charge of tau, which is consequently predicted to drift by <10% (+11 at pH 7.0 and +12 at pH 6.8). As the Coulombic interaction is essential in tau-RNA CC, such pH fluctuations can contribute to the variability we observed in the phase diagram.

**Author response image 1. respfig1:** pH changes of buffer. The pH values of 20 mM ammonium acetate and 20 mM HEPES were recorded upon addition of 20 μM tau or 60 μg/mL RNA.

Since we are currently unable to generate replicate data with smaller variability, we are working with a relatively high number of independent repeats to increase confidence in the tau-RNA phase diagram data. For example, if we take the same data as the reviewer took in Figure 2B, 5 independent measurements span 14 ºC to 20 ºC, with the average value constrained around 17 ºC and a standard deviation of ± 2 ºC. While not perfect, such data range was of sufficient quality to fit the temperature-dependent excluded volume parameter (FTS) or the temperature-dependent Flory-Huggins parameter (Voorn-Overbeek model) to a linear model and determine the location of the experimental data range within the theoretically determined phase diagram model.

To extract an experimental phase diagram with higher resolution, pH might have to be exceptionally tightly controlled, temperature tightly controlled and RNA degradation prevented, to name three possible factors that we can think of controlling in future measurements.

Besides variability, we also seriously considered the suggestion to determine equilibrium by sitting at a particular temperature and monitoring the turbidity values over time. By doing so, we have observed a steady decrease of turbidity which results from RNA degradation as written above. Even if turbidity is steady, it can result from absorbance of insoluble tau aggregates and thus cannot prove reversibility. Due to the technical challenges of controlling RNA degradation and the ambiguity of turbidity values we used spin labeling and electron paramagnetic resonance to accurately show tau is in a steady state, while using turbidity over temperature ramping along with microscopy to demonstrate tau-RNA CC is reversible and to estimate the phase separation conditions.

To address these comments, we added in the main text:

“We point out that there is certain level of variability in the observed T_cp_, which can result from pH fluctuation of ammonium acetate buffer upon tau/RNA addition, as well as RNA degradation as demonstrated in Figure 1—figure supplement 2.”

2) Subsection “Tau-RNA complex coacervate phase diagram”, last paragraph: The experimental results indicate that tau114 (134 residues, Figure 1—source data 1) has a higher propensity to undergo liquid-liquid phase separation (LLPS) with RNA than tau187 (207 residues). Can the authors suggest a plausible physical reason for this behavior? If one considers homopolymer LLPS or simple coacervation, propensity to LLPS tends to decrease with decreasing chain length, which is not being observed here for tau-RNA. Are there differences in sequence-specific features between tau114 and tau187 that may provide a plausible rationalization for the experimentally observed trend?

It is indeed true that there is a higher entropic penalty for shorter polymers to phase separate. This usually results in a lower salt concentration being sufficient to screen electrostatic interactions and dissolving the coacervate phase, as the reviewer correctly points out. This trend of decreased LLPS propensity with decreasing chain length has been observed in peptide/RNA experiments (see (1)). Although tau114 is shorter than tau187 (Figure 1—source data 1), it carries a similar net positive charge at pH ~ 7. If we assume a uniformly smeared out charge distribution, the average charge per amino acid is about twice as that of tau187 (Figure 2—source data 1). Consequently, under the same tau-RNA total mass concentration, tau114/RNA is expected to experience stronger tau-RNA Coulombic attraction, which may counteract the entropy penalty effect of the shorter chain.

Sequence-specific tau-RNA interaction is another plausible explanation. Experimental results have shown the 4R domain (K18) has a stronger binding affinity to RNA compared to full-length tau (2N4R) (see Zhang et al., 2017). Since tau114 is the N-terminal of tau187, largely corresponding to the 4R domain, it is possible that tau114 might experience stronger binding with RNA compared to tau187. However, experimentally, it is difficult to disentangle electrostatic Coulombic attractions from other enthalpic interactions. For simplicity we have not included any additional tau-RNA interactions into our theoretical and computational studies and model, but this could be included in the future, either in the mean-field (FH-VO) or in the FTS model through a Flory-Huggins interaction term acting between tau and RNA. We are currently developing such a model.

To address these comments, we have added the following:

“Comparison of the two constructs shows that tau114-RNA CC has a lower T_cp_ than tau187-RNA CC, suggesting it is more favorable to phase separation. […] Additional short-ranged sequence-specific interactions between tau114 and RNA that are not present in tau187 is another possibility that is not considered in the present model.”

3) In the authors' treatment using Flory-Huggin-Voorn-Overbeek (FH-VO, subsection “Flory-Huggins-Voorn-Overbeek Fit to Experimental Phase Diagram”), a temperature-dependent Flory chi parameter (Equation 1, see the aforementioned subsection) is used to fit the data and to produce the experimental observation that the tau-RNA LLPS tendency increases with increasing temperature (i.e., the system has a LCST). This phenomenon, which is also observed for elastin, is similar to cold denaturation of globular proteins, and its possible physical origin has been discussed recently in the context of biomolecular LLPS by Lin et al., 2018; see Figure 2C of this ref] in terms of the expected temperature-dependence of hydrophobic interactions. The strength of effective attraction between nonpolar groups tends to increase with increasing temperature, as has been demonstrated from model compound transfer experiments, a fact that has long been applied to theory of protein conformational transitions, see, e.g., Dill, Alonso and Hutchinson, 1989. The same trend has also been seen in explicit-water simulations of potentials of mean force of nonpolar solutes in water, see, e.g., Figure 4 in Dias and Chan, 2014. Beyond introducing the temperature-dependent chi parameter as a mere necessity for fitting data, it would be very helpful to the readers to include this perspective (and the above-cited references) in the authors' Discussion to provide a plausible physical rationalization for the observed trend.

We think the comparison of tau-RNA LLPS with cold denaturation of globular protein is inspiring. The contribution of effective attractions between nonpolar groups to the driving force in the tau-RNA LLPS process is very worthy of study. While the explicit testing of these effects is outside the scope of our paper, we have included such discussion in the main text and cited the above references. These new references are now Lin et al., 2018; Dias and Chan, 2014; Dill et al., 1989 in the main text.

“A temperature dependence of χ in the form of Equation 1 (consistent with the observed LCST), can originate from hydrophobic interactions between non-polar groups, whose interaction strength tends to increase with temperature (Lin et al., 2018; Dias and Chan, 2014). This explanation has also been used to describe cold denaturation of proteins (Dill et al., 1989).”

4) The statement "high crowding reagents (acting to lower the excluded volume)" is problematic. What do the authors mean by the statement? Could the current formulation of this statement be related to the aforementioned somewhat confused concept about excluded volume presented in the manuscript? Would it make more sense to say that crowding agents act to lower the chain configuration entropy of the dilute phase?

In explicit solvent, repulsive interactions between protein segments and crowding agents would lead to chain compaction in the dilute phase relative to a chain in pure solvent, in the absence of the crowding agent. Thus, on average, polymer chains will become more compact with increased crowding agents as the reviewer suggests. However, our model is an implicit solvent model, meaning that the solvent degrees of freedom are not explicitly included in the statistical mechanical description. The excluded volume potential is a pair potential of mean force, and is state dependent. As shown in (Jeon et al., 2016), mapping the explicit-crowder case onto an equivalent implicit-solvent case means that the solvent quality must change with increasing concentration of the crowders. This idea dates back to Flory’s ideality hypothesis and the effect of the crowding agents is to screen the bare excluded volume repulsion between protein segments. The effect of added crowding agents can be implicitly modeled by having an excluded volume strength parameter that decreases with increased concentration of crowding agents, implicitly accounting for this screening. This was what we meant by the statement “high crowding reagents (acting to lower the excluded volume).” The excluded volume parameter is a repulsive energy/volume between protein segments. This repulsive term causes segments to repel each other in order to maximize their exposure to the solvent in good solvent conditions. By lowering this repulsive strength parameter we are modeling the poorer solvent conditions such that the protein segments no longer swell as much as they would in the absence of crowding agents.

We have changed the manuscript to read:

“high crowding reagents (leading to solution conditions with a lower effective excluded volume parameter to model the poorer solvent environment in an implicit solvent model (Jeon et al., 2016))”.

5) In the field theoretic treatment (subsection “Field theoretic simulations of a coarse-grained model of tau-RNA complex coacervation”), the temperature dependent attraction in the form of (1-theta/T) is presented as a temperature-dependent excluded-volume potential. Although it may be mathematically convenient to introduce temperature dependence as a modified "excluded volume" term, doing so is physically inaccurate and conceptually misleading. Excluded-volume interaction is a hard-core repulsion. Inter-chemical-group attractive interactions such as hydrophobic effect do not abolish the hard cores of the atoms. They just affect the potential energy landscape close to but beyond the hard cores. With this in mind, while the usage of a single Gaussian function for this "excluded volume" energy may be a mathematical simplification that is necessary for field theoretic formulations [the Γ(r) function in the last paragraph of the subsection “Coarse grained polyelectrolyte model used in FTS”], it is important to recognize that it is not entirely physically accurate (e.g., one does not use such a functional form in most explicit-chain molecular dynamics force fields). The reason is that in the authors' set up the molecular hard cores can be penetrated but in reality they cannot, and this can have consequences on the predictions of volume fractions, etc. Casting the solvent-mediated attraction as a reduction in excluded volume is also conceptually inconsistent with FH-VO because the FH excluded volume is approximately accounted for by the mixing entropy term, and the introduction of the chi parameter does not envision penetrating any excluded volume. Although this point is largely conceptual in the present context and the reviewers do not suggest redoing the field theory simulation using a more realistic excluded volume potential, it is still importance to have conceptual clarity in the authors' Discussion so as to avoid, e.g., oxymoronic statements such as “the excluded volume becomes attractive”. With a recognition of this limitation, the authors may also wish to qualify their statement regarding the "approximation-free" nature of their simulations (it may be approximation-free with regard to the assumed model potential but the model potential itself is limited at least as described above).

The temperature dependence of the excluded volume potential originates from our use of an implicit solvent model; therefore, the excluded volume potential is a potential of mean force, and thus is state-dependent (including on temperature). The potential represents in an average way the effect of the solvent on the chain configurations, and was introduced by Flory to account for chain swelling of a polymer in solution. For example, setting the excluded volume to zero does *not imply* as the review suggests that we “abolish the hard core-atoms,” but is a limiting case where the net interactions between the solvent molecules (not represented) and polymer chains (including all hard-core repulsive interactions) exactly counter-act each other. This is the so-called theta temperature. The temperature dependence of the “effective” excluded volume parameter is well-documented and dates back to Flory. See for example reference (2) or (3).

This temperature-dependent excluded volume has nothing to do with the Gaussian-smearing, and the result would be the same when using a delta-function for the excluded volume term. In general, for an implicit solvent model, where we have integrated out the solvent degrees of freedom, the excluded volume will *always* be temperature dependent. A similar situation occurs in the Chandler-Weeks-Andersen (CWA) liquid-state theory when the effective hard-sphere diameter is found by equating the compressibility of a reference system to the compressibility of a hard sphere system, giving an effective hard sphere diameter which is both temperature and density dependent.

In our particular case, the repulsive strength decreases as temperature increases. This is simply stating that as the temperature *increases* the system approaches the theta condition, which is in agreement with experimental observation (4). Thus, it is not “oxymoronic” to state that the “excluded volume becomes attractive.” The conditions under which the potential becomes attractive (negative values of the excluded volume), indicate the presence of effective attractive forces between protein segments that lead to chain collapse due to the preference of solvent-solvent interactions over solvent-monomer interactions. This would be in agreement with the large literature on hydrophobic collapse, which can be presented as a function of the solvent quality.

To explain this temperature dependence explicitly, we have included the following:

“The excluded volume parameter *v* can be related to the residue-residue non-Coulombic interaction potential U(r) as

ν=∫1-e-UrkBTd3r

and is …”

In line with the reviewer’s suggestion, we qualify our statements regarding the approximation-free nature of the simulations. They are exact with regard to the polymer physics model, but the model is itself a coarse-grained simplification of the real-system.

We have replaced “serves as an approximation free method” with

“FTS is a numerical approach that allows one to fully account for fluctuations, and thus to compute equilibrium properties from a suitably chosen coarse-grained representation of the true system without the need for analytical approximation.”

And we have changed “approximation-free” to “complete.”

6) Discussion, third paragraph, schematic equation: It would be much clearer to view the overall attractive interactions among tau proteins and RNA as an "effective free energy of interaction" that contains both enthalpic and entropic parts (wherein the entropic contribution comes largely from solvent entropy), and this effective interaction free energy [which, by the way, correspond to the vertical axis in energy landscape drawing for protein folding] is then balanced by configurational/conformational entropy of the chains (which, in the FH-VO approximation is given by the classical mixing term). The authors may wish to adopt such a narrative.

We thank the reviewer for proposing the viewpoint of “effective free energy of interaction”. In fact, the equation does include such a narrative (see below).

ΔGCC=ΔHtau/RNA⏟(-)-TΔScomb⏟(-)+ΔHex-TΔSnoncomb⏟excludedvolumeorχ

In this equation, ΔGCCis the effective free energy of tau-RNA liquid-liquid phase separation (complex coacervation, CC). ΔHtau/RNAis the enthalpic contribution resulting from attractive interactions among tau proteins and RNA, corresponding to the enthalpy change in protein folding. TΔScombis the combinatoric entropy part, which in the FH-VO approximation is given by the classical mixing term (∑ϕiNilnϕi, as in Equation S1).

Our results showed that only using the first two terms can NOT capture the temperature-dependent phase separation behavior of tau-RNA (i.e. LCST). Therefore, we introduce the additional enthalpic and entropic terms ΔHex and TΔSnoncombin order to capture the non-ideal behavior.

7) More discussions about controls relevant to their FTS simulations is needed to evaluate the FTS methodology.

The idea of describing a polymer chain in solution with functional integrals dates back to the pioneering work of Edwards, de Gennes, and Helfand in the 1970s. We employ a numerical technique called Complex-Langevin (CL) sampling to solve for the ensemble-averaged observables. This technique (see (5)) has been used to study synthetic polymer systems in dozens of papers. In fact, FTS via CL sampling was introduced ten years ago to study polyelectrolyte complex coacervation, although the full phase diagrams could not be obtained yet at that time (Lee, Popov and Fredrickson, 2008; Popov, Lee and Fredrickson, 2007).

More recent work provides the first phase diagram from FTS via CL sampling for simple polyelectrolytes in 2017 (Delaney and Fredrickson, 2017). This work can be seen as the benchmark of our current implementation. The novelty here lies in applying this technique that is well-established in the polymer community (Delaney and Fredrickson, 2016) to a biological system, namely of tau-RNA CC, and mimicking biological conditions.

We have added the following to the main text:

“Field theoretic simulation (FTS) has been widely used to model synthetic polymers (Delaney and Fredrickson, 2016), including LLPS in polyelectrolytes and polyampholytes (Delaney and Fredrickson, 2017; Lee, Popov and Fredrickson, 2008; Popov, Lee and Fredrickson, 2007). […] The main advantage of this approach is that it does not rely on analytical approximations, and thus can be useful when comparing directly to experiment at conditions where such approximate theories break down”

8) When the authors compared simulation to experiment (subsection “Comparison between simulation and experiment”), how was the Bjerrum length (l_B_) parameter obtained from Equation 3 (there is no Equation 3 in the manuscript)? How Equation 2 can be used to obtain l_B_ if the relative dielectric constant term still needs to be determined? This is a very important section of the paper and a more detailed discussion is needed here.

We thank the reviewer for pointing out this important point. Comparison to experiment requires estimating a value for the parameters in the polymer model. Given the “coarseness” of the model and for simplicity, we have set ε_r_ = 80 and b = 0.4 nm for all cases. The Bjerrum length (*l_B_*) was then calculated according to Equation 2. The reviewer is correct that in reality the dielectric constant is temperature dependent. For example, from (Malmberg and Maryott,1956), the dielectric constant of water at 10 °C is 83.8 and at 30 °C is 76.5. These values will change yet again slightly in different buffer conditions. We have ignored theses complexities and estimated ε_r_ as ~ 80. Our results are best interpreted as showing the relative contributions from the electrostatics vs. the excluded volume effects, both of which are important. One could choose a more detailed parameterization of the FTS simulations, but since the excluded volume parameter is unknown and important, the quality of the results would be similar.

We have added the following statement:

“Although ε_r_ will depend on temperature, for simplicity we treat this parameteras a constant such that the Bjerrum length *l_B_*~ 1/T. Thus, *l_B_* can be estimated at the experimental cloud point temperature directly from Equation 2 (Figure 6A), which leaves only one unknown parameter *v*.”

9) In the conclusion, the authors state that the simulation provides "knowledge that can be used to design experiments that perturb this parameter space in vivo". It would be extremely useful if the authors test FTS and their results by considering tau phosphorylation. Even just modeling tau phosphorylation without RNA would be useful. Additional discussion along this line, if attainable in a timely fashion, is preferred, but not required by the reviewers and the editors.

The topic of post-translational modification and phosphorylation affecting liquid-liquid phase separation is what we are currently investigating. However, given the interest in how phosphorylation might affect tau-RNA CC, we have included a brief preview in the main text which is copied here:

“To demonstrate that FTS of tau-RNA LLPS can be applied to consider the effect of post translational charge modification of tau, we tested FTS of phosphorylation as an example. […] In full-length tau, most of the phosphorylated sites are not in the positively charged repeat domain region, indicating that its LLPS might follow the same principles as the self-coacervation seen in polyampholytes (Delaney and Fredrickson, 2017).”

10) In Figure 6C, for the 120mM NaCl data, the simulated transition temperatures (open green squares) have a significantly higher rate of decrease with respect to increasing [tau] than the corresponding experimental data (blue circles). Could the authors provide a plausible reason for this discrepancy? The Bjerrum length l_B_ is plotted in Figure 6A. Could the authors confirm in the figure legend that this is just a plot of Equation 2 and no temperature dependence of the relative permittivity ε_r_ was introduced in Figure 6A?

We confirm as stated above that no explicit temperature dependence was included in the relative permittivity, and yes, Figure 6A is the strictly linear plot according to Equation 2 with ϵr=80 plotted for clarity, whereas Figure 6B is a linear fit to the data to determine the excluded volume repulsive potential parameter as a function of temperature. This is made clearer in the first paragraph of the subsection “Comparison between simulation and experiment”.

We find it difficult to infer with confidence a difference in the rate of decrease between experiment and simulation, given the limited experimental data (3 data points). Nonetheless, we find it promising and quite remarkable that without introducing any new fit parameters, the simulation (upon adding salt) and the experiment agree reasonably well in estimating the dilute phase concentration and how it changes with salt concentration. This is a positive result, and not something that one would expect using a more approximate theory.

References:

1) Aumiller and Keating, Nature Chemistry, 8, 2016, p. 129

2) Rubinstein and Colby, Polymer Physics, Oxford University Press, Section 3.3.3

3) Gert Strobl, The Physics of Polymers, 3rd ed., Springer, Section 2.3

4) Bianconi, Ciasca, Tenenbaum, Battisti, Campi, J Biol Phys. 2012 Jan 1;38(1):169–79

5) Fredrickson, Ganesan, and Drolet,Macromolecules 2002, 35, 16–39